# FedQueue: Queue-Aware Federated Learning for Cross-Facility HPC Training

**Yijiang Li** [1]  **Emon Dey** [1]  **Zilinghan Li** [2]  **Krishnan Raghavan** [1]  **Ravi Madduri** [2]  **Kibaek Kim** [1]

## Abstract

Federated learning (FL) across multiple HPC facilities faces stochastic *admission delays* from batch schedulers that dominate wall-clock time. Synchronous FL suffers from severe stragglers, while asynchronous FL accumulates stale updates when queues spike. We propose FEDQUEUE, a queue-aware FL protocol that incorporates scheduler delays directly into training and aggregation, which (i) predicts per-facility queue delays online to budget local work, (ii) applies cutoff-based admission that buffers late arrivals to bound staleness, and (iii) performs staleness-aware aggregation to stabilize heterogeneous local workloads. We prove the convergence for non-convex objectives at rate $\mathcal{O}(1/\sqrt{R})$ under bounded staleness, and show that the admission controls yield bounded staleness with high probability under queue-prediction error. Real-world cross-facility deployment of FEDQUEUE shows 20.5% improvement over baseline algorithms. Controlled queue simulations demonstrate robust improvement over the baselines; in particular, up to 60% reduction in time to reach a target accuracy level under high queue variance and non-IID partitions.

## 1. Introduction

Federated learning (FL) enables collaborative model training across data silos without centralizing raw data (McMahan et al., 2017; Kairouz et al., 2021). An important application is *cross-facility* FL for scientific and industrial workloads (Zhang et al., 2025; Zhuang et al., 2025; Kim et al., 2024; Li et al., 2026), where each client is a high-performance computing (HPC) facility operating under a batch scheduler (e.g., Slurm, PBS). Such settings are increasingly important for training large scientific foundation models, where no single facility holds sufficient data or compute and institutional or regulatory constraints preclude centralizing raw data. The dominant challenge in this application is *admission delay*: jobs may wait minutes to hours in scheduler queues before starting, with wait times varying unpredictably due to system load and priority policies (Feitelson, 2015).

Queue-induced delays fundamentally change FL dynamics. Synchronous protocols like FedAvg (McMahan et al., 2017) suffer severe straggler effects as the slowest queued facility dictates progress. Fully asynchronous protocols (Xie et al., 2020; Xu et al., 2024) avoid blocking but accumulate heavy-tailed *staleness* when delays are stochastic, aggregating updates from outdated models. Existing system-aware methods (Li et al., 2024) profile compute throughput but assume static availability patterns and do not model or control scheduler-driven queue dynamics.

We propose FEDQUEUE, a queue-aware FL protocol that *predicts and controls scheduler admission delays* to achieve robust wall-clock progress with *provably bounded staleness*. Our focus is on systems heterogeneity arising from scheduler-induced admission delay. FEDQUEUE combines: (i) *online queue prediction* to budget per-facility job time and local steps, (ii) *cutoff-based admission* that avoids blocking while bounding staleness, and (iii) *staleness-aware aggregation* with inverse learning rate scaling. We prove non-convex convergence under provably bounded staleness for FEDQUEUE. These guarantees ensure that FEDQUEUE converges reliably despite queue variability, a key requirement for production HPC deployments where queue dynamics are unpredictable and time-varying.

The key contributions of this work include: **(1) Queue-aware FL algorithm.** We formalize cross-facility FL under stochastic scheduler delays as a first-class signal, and introduce FEDQUEUE, combining online prediction, adaptive budgeting, admission control, and staleness-aware aggregation into a principled framework. **(2) Convergence theory with staleness control.** We prove convergence at rate $\mathcal{O}(1/\sqrt{R})$ with bounded staleness, which our admission policy achieves with high probability under sub-Gaussian prediction errors. **(3) Empirical validation.** Real-world deployment shows 20.5% improved loss over the baselines while experiments with controlled synthetic queues show up to 60% faster convergence and validate our theory.

[1]Mathematics and Computer Science Division, Argonne National Laboratory, Lemont, IL, USA [2]Data Science and Learning Division, Argonne National Laboratory, Lemont, IL, USA. Correspondence to: Kibaek Kim <kimk@anl.gov>.

*Proceedings of the 43rd International Conference on Machine Learning*, Seoul, South Korea. PMLR 306, 2026. Copyright 2026 by the author(s).

## 2. Related Work

### 2.1. FL Under Asynchrony and Staleness

FedAvg (McMahan et al., 2017) and many follow-ups (Li et al., 2020; Karimireddy et al., 2020) assume synchronous rounds, which are sensitive to stragglers. Asynchronous and buffered variants reduce blocking by aggregating updates as they arrive (Xie et al., 2020; Xu et al., 2024; Nguyen et al., 2022). Their analyses typically require bounded staleness (or impose staleness-dependent weighting) to control instability in non-convex settings (Nguyen et al., 2022; Fraboni et al., 2023; Forootani & Iervolino, 2025). A complementary line studies semi-/deadline-asynchronous protocols that explicitly control delay via deadlines or stale-update filtering (Yu et al., 2023; 2024b; Liu et al., 2024; Damaskinos et al., 2022). More recently, FADAS (Wang et al., 2024) integrates asynchrony into adaptive federated optimization and proposes a delay-adaptive learning-rate strategy that down-scales the global step size when client delays exceed a threshold. It operates as a reactive server-side optimizer adjustment. FEDQUEUE targets a distinct delay mechanism—*scheduler-induced admission delay*—and enforces bounded staleness via queue prediction and admission control rather than assuming it a priori or reacting to delay after the fact.

### 2.2. System Heterogeneity and Client Availability

Beyond asynchrony, FL must cope with heterogeneous data and systems. Methods such as FedProx and SCAFFOLD stabilize training under non-IID data (Li et al., 2020; Karimireddy et al., 2020). Other approaches adapt client selection or resource allocation under heterogeneous compute and participation (Ribero et al., 2022; Wang & Ji, 2022; Garg et al., 2025). FedCompass (Li et al., 2024) is closely related: it profiles client throughput and allocates local steps to equalize progress. However, throughput profiling alone does not capture *time-varying admission delay* that can dominate wall-clock time in batch-scheduled HPC. FEDQUEUE complements profiling-based approaches by treating admission delay as a critical time-varying signal and shaping the realized staleness distribution through admission control.

### 2.3. HPC Scheduling and Queue-Time Prediction

Batch schedulers such as Slurm, PBS, and Maui govern most HPC systems (Yoo et al., 2003; Henderson, 1995; Jackson et al., 2001), and extensive empirical work characterizes their workload dynamics and waiting-time variability (Feitelson, 2015). Queue wait-time prediction has been studied to support workflow planning and scheduling decisions (Brown et al., 2022; 2024; Jancauskas et al., 2019). We use a lightweight and easy to deploy predictor, exponential weighted moving average (EWMA) across sites, but our analysis is prediction-agnostic.

### 2.4. Cross-Facility Training on HPC

Cross-facility FL frameworks (e.g., xFFL) demonstrate end-to-end orchestration across supercomputers using workflow systems and remote job submission (Casella, 2025; Colonnelli et al., 2024; Li et al., 2020). These efforts surface practical issues such as queue variability and robustness, but focus on orchestration rather than algorithm design with queue-aware control and convergence guarantees. FEDQUEUE fills this gap by providing a queue-aware FL protocol with theory and time-to-quality evaluation under real scheduler dynamics.

### 2.5. Positioning

Most asynchronous/buffered FL methods treat delay and staleness as exogenous and analyze convergence under an assumed staleness bound (Xie et al., 2020; Nguyen et al., 2022). Profiling-based system-aware methods, in contrast, focus on stable throughput heterogeneity (Li et al., 2024). FEDQUEUE explicitly models scheduler-driven admission delay, predicting it online, and using cutoff-based admission to *induce* bounded staleness with high probability, enabling principled convergence guarantees.

## 3. Problem Formulation

### 3.1. Problem Setup

We consider FL over $K$ clients (HPC facilities), indexed by $k \in \{1, \dots, K\}$. Client $k$ holds a local dataset $\mathcal{D}_k$ and minimizes a local empirical risk $F_k(w)$. The global objective is $\min_{w \in \mathbb{R}^p} \quad F(w) := \sum_{k=1}^{K} p_k F_k(w)$, where $w \in \mathbb{R}^p$ is the model parameter vector, $p_k \geq 0$ are client weights with $\sum_k p_k = 1$ (typically $p_k \propto |\mathcal{D}_k|$). We are motivated by the *cross-facility* regime where each client is a batch-scheduled HPC system and the dominant source of systems heterogeneity is *scheduler-induced admission delay* (Feitelson, 2015). Importantly, scheduler-induced admission delay does not modify the learning objective $F(w)$; rather, it changes when each client can compute and return an update. As a result, the server may apply an update computed from a stale global model rather than the current iterate. This timing-induced staleness is the primary mechanism by which queue delays affect convergence and wall-clock time-to-quality.

Unlike static throughput heterogeneity, queue delays in HPC systems are stochastic and time-varying due to batch scheduling policies and changing system load (Feitelson, 2015). Critically, these delays are *not known in advance* and thus the server must predict them to effectively coordinate training. This uncertainty necessitates an adaptive approach that combines queue prediction, dynamic work budgeting, and staleness-aware aggregation.

## 3.2. Round Timeline and Arrival Times

We specialize the standard asynchronous-FL timeline notation to batch-scheduled HPC settings. Throughout, superscripts $r$ index server rounds (out of total $R$ rounds). To compare methods under stochastic scheduler delays, we parameterize training by a server wall-clock schedule where the server operates on a fixed timer, advancing a logical round counter $r$ every $T_{\text{sync}}$ seconds (a user-chosen *synchronization horizon*), independent of which updates have arrived. This fixed cadence defines deadlines for update admission.

At the beginning of round $r$ (at time $rT_{\text{sync}}$), the server broadcasts the current global model $w^{(r)}$ and triggers job submission at each facility. Client $k$ experiences a queue (admission) delay $q_k^{(r)} \geq 0$ before the job starts, then trains for local compute time $h_k^{(r)} \geq 0$, and returns an update at arrival time $a_k^{(r)} := rT_{\text{sync}} + q_k^{(r)} + h_k^{(r)}$. Because $q_k^{(r)}$ is stochastic and unknown at round start, updates may arrive after the server has already advanced to subsequent rounds.

## 3.3. Staleness and Update Buffering

Since jobs may complete after the server has advanced the global model, updates can be computed on stale parameters. Let $R_k^{(r)} \leq r$ denote the index of the global model that client $k$ used to compute an update that is *aggregated* at server round $r$, the staleness is $\tau_k^{(r)} := r - R_k^{(r)} \geq 0$. For example, if client $k$ receives $w^{(s)}$ at the start of round $s$ but its update arrives after round $s$ has ended, that update will be buffered and aggregated at a later round $r > s$ with staleness $\tau_k^{(r)} = r - s$.

Excessive staleness degrades convergence because the update direction computed from $w^{(R_k^{(r)})}$ may no longer align well with the current model $w^{(r)}$. To control staleness while maintaining high system utilization, we need mechanisms to predict queue delays and adaptively budget local work.

# 4. Queue-Aware FL Algorithm

In this section, we present our Queue-Aware FL (FEDQUEUE) algorithm, which addresses the challenges of training across multiple HPC facilities with unpredictable queue delays by combining three key components: **(1) Queue prediction and adaptive budgeting:** Maintain online predictions of queue delays and dynamically allocate per-client compute budgets to target consistent wall-clock progress. **(2) Deadline-based admission control:** Use cutoff times to bound update staleness while buffering late arrivals for incorporation in subsequent rounds. **(3) Staleness-aware aggregation:** Down-weight stale updates and scale learning rates inversely with local step counts to stabilize optimization.

## 4.1. Server Algorithm Components

Algorithm 1 presents the server-side protocol for FEDQUEUE. At the start of each round $r$, the server maintains queue delay predictions $\hat{q}_k^{(r)}$ for all clients, computes per-client job-time budgets $J_k^{(r)}$ and local step budgets $E_k^{(r)}$ based on the profiled throughput $c_k$ and predictor state $\hat{q}_k^{(r)}$, determines adaptive learning rates $\eta_k^{(r)}$, and broadcasts $(w^{(r)}, r, J_k^{(r)}, E_k^{(r)}, \eta_k^{(r)})$ to all clients. The server then collects arriving updates until a deadline, aggregates all available updates with staleness-aware weighting, and buffers any late arrivals for future rounds.

---

**Algorithm 1** FEDQUEUE Server

---

**input** Initial model $w^{(0)}$, synchronization horizon $T_{\text{sync}}$, safety buffer $\delta$, EWMA rate $\alpha$, base learning rate $\eta_{\text{base}}$, staleness decay $\phi(\cdot)$, facility weights $\{p_k\}_{k=1}^{K}$, throughputs $\{c_k\}_{k=1}^{K}$, maximum rounds $R_{\max}$.

**output** Global models $\{w^{(r)}\}_{r=0}^{R_{\max}}$.

1: Initialize queue predictions $\hat{q}_k$ for all $k$; buffer $\mathcal{B} \leftarrow \emptyset$; $\mathcal{A}^{(-1)} \leftarrow \{1, \ldots, K\}$
2: **for** $r = 0, 1, \ldots, R_{\max} - 1$ **do**
3:     **for** $k \in \mathcal{A}^{(r-1)}$ **do**
4:         $J_k^{(r)} \leftarrow T_{\text{sync}} - \hat{q}_k - \delta$;    $E_k^{(r)} \leftarrow \lfloor c_k J_k^{(r)} \rfloor$
5:     **end for**
6:     $E_{\min}^{(r)} \leftarrow \min_k E_k^{(r)}$
7:     **for** $k = 1, \ldots, K$ **do**
8:         $\eta_k^{(r)} \leftarrow \eta_{\text{base}} E_{\min}^{(r)} / E_k^{(r)}$
9:         Broadcast $(w^{(r)}, r, J_k^{(r)}, E_k^{(r)}, \eta_k^{(r)})$ to facility $k$
10:     **end for**
11:     Set cutoff time $t_{\text{cut}}^{(r)} \leftarrow (r+1) T_{\text{sync}}$
12:     Receive messages $(k, s, \Delta, q)$ at time $t$:
13:         $\mathcal{B} \leftarrow \mathcal{B} \cup \{(k, s, \Delta, t)\}$,    $\hat{q}_k \leftarrow (1 - \alpha)\hat{q}_k + \alpha q$
14:     $\mathcal{A}^{(r)} \leftarrow \{(k, s, \Delta) : (k, s, \Delta, t) \in \mathcal{B}, \ t \leq t_{\text{cut}}^{(r)}\}$
15:     $\mathcal{B} \leftarrow \mathcal{B} \setminus \{(k, s, \Delta, t) \in \mathcal{B} : t \leq t_{\text{cut}}^{(r)}\}$
16:     $S^{(r)} \leftarrow \sum_{(k,s,\Delta) \in \mathcal{A}^{(r)}} p_k \phi(r - s)$
17:     $w^{(r+1)} \leftarrow w^{(r)} + \frac{1}{S^{(r)}} \sum_{(k,s,\Delta) \in \mathcal{A}^{(r)}} p_k \phi(r - s) \Delta$
18: **end for**

---

### 4.1.1. QUEUE DELAY PREDICTION

Queue delays $q_k^{(r)}$ are not known when planning round $r$, the server must maintain an online predictor $\hat{q}_k^{(r)}$. When client $k$'s update arrives, the server observes the realized queue delay $q_k^{(r)}$ and updates its predictor for future rounds. We employ EWMA, a lightweight baseline commonly used in queue-time prediction (Jancauskas et al., 2019; Brown et al., 2022; 2024): $\hat{q}_k^{(r+1)} = (1 - \alpha)\hat{q}_k^{(r)} + \alpha q_k^{(r)}$, where $\alpha \in (0, 1)$ is the EWMA rate controlling the trade-off between responsiveness to recent observations and noise stability.

Our convergence analysis does not assume a particular predictor form (prediction-agnostic framework): it only re-

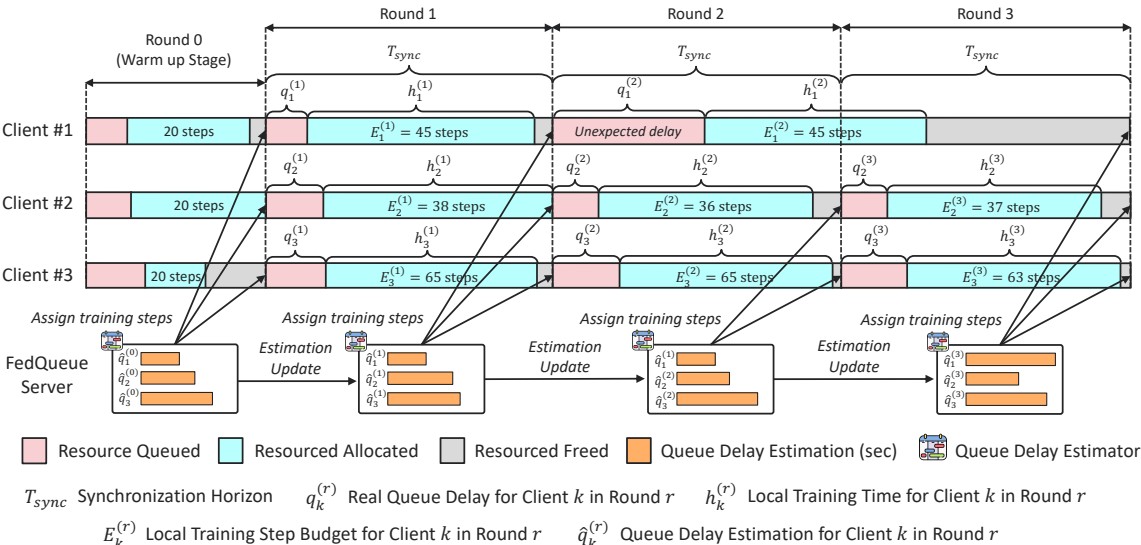

*Figure 1.* **Illustration of the FEDQUEUE Algorithm.** The FEDQUEUE server first obtains initial queuing delay estimates for each client $(\hat{q}_k^{(0)})$ during a warm-up stage, and accordingly assigns the number of local training steps $(E_k^{(r)})$. In each round $r$, the FEDQUEUE server updates the estimates based on the recent queuing delay $(q_k^{(r)})$ and performs global aggregation using all client updates received before the admission deadline $(rT_{\text{sync}})$. If a certain client fails to return its update within the deadline due to an unexpected delay (e.g., Client 1 in Round 2), its update will be deferred and incorporated in the next aggregation round, with the staleness factor applied.

quires that prediction errors satisfy a sub-Gaussian concentration property (see Section 5). This allows practitioners to substitute more sophisticated predictors directly as needed for their specific HPC environments.

### 4.1.2. ADAPTIVE WORK BUDGETING

Given the queue delay prediction $\hat{q}_k^{(r)}$, the server allocates a per-round job-time budget $J_k^{(r)}$ to each client to target completion within the synchronization horizon $T_{\text{sync}}$ (Line 4 in Algorithm 1): $J_k^{(r)} := T_{\text{sync}} - \hat{q}_k^{(r)} - \delta$, where $\delta > 0$ is a safety buffer that accounts for prediction error and system overheads (e.g., data transfer, job initialization). The corresponding local training step budget is $E_k^{(r)} := \left\lfloor c_k J_k^{(r)} \right\rfloor$, where $c_k > 0$ is the profiled training throughput of client $k$ in local SGD steps per second. We therefore enforce that local training time $h_k^{(r)}$ respects the job-time budget for all clients (i.e., $h_k^{(r)} \leq J_k^{(r)}$ for all $k, r$).

This adaptive budgeting mechanism enables FEDQUEUE to balance heterogeneous queue conditions: clients experiencing longer predicted queues receive smaller compute budgets to increase their likelihood of meeting the round deadline, while clients with shorter predicted queues can perform more local work.

### 4.1.3. ADAPTIVE LEARNING RATE SCALING

The adaptive budgeting mechanism produces heterogeneous local step counts $E_k^{(r)}$ across facilities due to varying queue predictions and compute speeds. Without compensation,

clients performing more local steps would dominate the aggregation and cause excessive drift from the global model.

To stabilize training under heterogeneous local step counts $E_k^{(r)}$, we adopt an inverse learning rate scaling strategy inspired by FedCompass (Li et al., 2024): $\eta_k^{(r)} = \eta_{\text{base}} E_{\min}^{(r)}/E_k^{(r)}$ (Line 8 in Algorithm 1). This scaling equalizes the effective local displacement across heterogeneous step budgets. Consider the displacement of client $k$'s parameters after $E_k^{(r)}$ local SGD steps, to first order, the cumulative displacement is approximately $\eta_k^{(r)} \cdot E_k^{(r)} \cdot \|\nabla F_k(w^{(r)})\|$. Without compensation (i.e., $\eta_k^{(r)} = \eta_{\text{base}}$), this quantity scales with $E_k^{(r)}$, so clients with larger budgets accumulate disproportionately large updates and dominate the aggregation. Substituting $\eta_k^{(r)} = \eta_{\text{base}} E_{\min}^{(r)}/E_k^{(r)}$ cancels the $E_k^{(r)}$ factor, yielding an effective displacement of approximately $\eta_{\text{base}} \cdot E_{\min}^{(r)} \cdot \|\nabla F_k(w^{(r)})\|$—independent of each client's local step budget. This relationship is used explicitly in our analysis (Section 5) to bound the accumulated local drift.

### 4.1.4. DEADLINE-BASED ADMISSION CONTROL

To bound staleness while maintaining high utilization, FEDQUEUE uses a deadline-based admission policy with buffering (Lines 11-17 in Algorithm 1). At the end of each round $r$, the server sets a cutoff time $t_{\text{cut}}^{(r)} := (r+1)T_{\text{sync}}$ and aggregates all updates that have arrived by this deadline. For a job submitted at the start of round $s$ (time $sT_{\text{sync}}$), the update is included in the aggregation for round $s$ if and only if $a_k^{(s)} \leq (s+1)T_{\text{sync}}$. Updates arriving after the

cutoff are not discarded; instead, they are buffered (Line 15 adds updates to $\mathcal{B}$) and incorporated in the first subsequent round whose cutoff they meet. Specifically, an update produced from $w^{(s)}$ and arriving at time $a_k^{(s)}$ is aggregated at next available round, i.e., round $r$ such that

$$r = \min\{j \geq s : a_k^{(s)} \leq (j+1)T_{\text{sync}}\}$$

with staleness $\tau_k^{(r)} = r - s$. This buffering mechanism ensures that no computational work is wasted while still maintaining bounded staleness through the cutoff discipline.

#### 4.1.5. STALENESS-AWARE AGGREGATION

To mitigate the impact of stale updates while retaining the benefits of asynchrony, we down-weight delayed updates using a staleness decay function $\phi : \mathbb{N} \to \mathbb{R}_+$ with $\phi(0) = 1$ (Lines 16-17 in Algorithm 1). We use harmonic decay by default (with a parameter $\beta$), $\phi(\tau) = (1 + \beta\tau)^{-1}$. A sensitivity study over values of $\beta$ and a comparison with exponential decay are provided in Appendix C.

The global model is updated using the staleness-weighted aggregation:

$$w^{(r+1)} = w^{(r)} + \frac{1}{S^{(r)}} \sum_{k \in \mathcal{A}^{(r)}} p_k \phi(\tau_k^{(r)}) \Delta_k^{(r)},$$

where $\mathcal{A}^{(r)}$ denotes the set of updates meeting the cutoff deadline for round $r$ (Line 14), $\Delta_k^{(r)}$ denotes the update from the client, and $S^{(r)} := \sum_{k \in \mathcal{A}^{(r)}} p_k \phi(\tau_k^{(r)})$ is a normalization factor ensuring proper scaling (Line 16).

#### 4.1.6. PRACTICAL PARAMETER SELECTION

FEDQUEUE's parameters can be guided by HPC system characteristics rather than fine-tuned. The synchronization horizon $T_{\text{sync}}$ plays the role of a round duration and can be set from pre-deployment scaling studies that profile nominal throughput $c_k$ and typical queue waits; the warm-up stage then initializes $\hat{q}_k$ so adaptive budgeting starts on realistic values. The safety buffer $\delta$ need not be precisely tuned, but should be set conservatively to keep most clients within $T_{\text{sync}}$, trading a small amount of local progress for tighter staleness concentration, as characterized in Lemma 5.4 and Figure 4. For the EWMA rate $\alpha$ a moderate value of 0.5 keeps client arrivals within $T_{\text{sync}}$ in our sweep, while both slower ($\alpha = 0.1$) and more aggressive ($\alpha = 1.0$) tracking induce late arrivals as shown in Figure 8 and Appendix E.2. The staleness-decay parameter $\beta$ should likewise be chosen conservatively; our sensitivity study in Appendix C shows performance is stable across $\beta \in \{0.25, 0.5, 1.0\}$ for both harmonic and exponential decay, with harmonic $\beta = 0.5$ as the recommended default. Finally, because scheduled maintenance windows are typically announced in advance, practitioners can avoid initiating training during anticipated disruptions.

### 4.2. Client Algorithm Components

---

**Algorithm 2** FEDQUEUE Client (Facility $k$)

---

**input** $(w^{(r)}, r, J_k^{(r)}, E_k^{(r)}, \eta_k^{(r)})$ from server.
**output** Message $(k, r, \Delta_k^{(r)}, q_k^{(r)})$.
1: $t_{\text{sub}} \leftarrow$ current time; submit a job for $J_k^{(r)}$ seconds.
2: Set $t_{\text{start}} \leftarrow$ start time and $q_k^{(r)} \leftarrow t_{\text{start}} - t_{\text{sub}}$.
3: Initialize $w_k \leftarrow w^{(r)}$; run local SGD for at most $E_k^{(r)}$ steps (or until time budget expires) with step size $\eta_k^{(r)}$.
4: $\Delta_k^{(r)} \leftarrow w_k - w^{(r)}$.
5: Send $(k, r, \Delta_k^{(r)}, q_k^{(r)})$ to server.

---

Algorithm 2 presents the client-side protocol for FEDQUEUE. Once each client $k$ receives the broadcast from the server, it submits a batch job to its local scheduler, observes the realized queue delay $q_k^{(r)}$, runs up to $E_k^{(r)}$ local SGD steps (or stops when the job-time budget expires), and returns the update $\Delta_k^{(r)}$ (Line 4) along with the round index $r$ and the observed queue delay to the server for determining staleness and updating queuing delay estimates.

Figure 1 illustrates the lifecycle of an FL experiment under the FEDQUEUE algorithm. In practice, we implement an extra warm-up stage to obtain the initial queuing estimates.

## 5. Theoretical Analysis

This section establishes theoretical convergence guarantees for FEDQUEUE under standard non-convex assumptions. We prove that its convergence rates are comparable to existing asynchronous FL methods while explicitly accounting for queue delay prediction error and staleness variance.

To establish convergence guarantees, we make the following standard assumptions in non-convex federated optimization.
**Assumption 5.1** (L-smooth objective). The global objective $F$ is $L$-smooth, i.e., for all $w, w' \in \mathbb{R}^p$:

$$\|\nabla F(w) - \nabla F(w')\| \leq L\|w - w'\|.$$

**Assumption 5.2** (Bounded Gradient Variance). The stochastic gradient has bounded variance. For each facility $k$ and any $w \in \mathbb{R}^p$:

$$\mathbb{E}_{\xi \sim \mathcal{D}_k} \left[ \|\nabla_\xi \ell(w; \xi) - \nabla F_k(w)\|^2 \right] \leq \sigma_k^2,$$

where $\sigma_k^2$ is the local gradient variance at facility $k$.

**Assumption 5.3** (Bounded Dissimilarity). The local objectives exhibit bounded heterogeneity. There exists $G \geq 0$ such that for all $k \in [K]$:

$$\|\nabla F_k(w) - \nabla F(w)\|^2 \leq G^2.$$

Assumptions 5.1 and 5.2 are standard in non-convex federated optimization and hold for typical neural network

training with smooth losses such as cross-entropy or mean squared error.

Assumption 5.3 is a common tool in the analysis of heterogeneous FL and appears in SCAFFOLD (Karimireddy et al., 2020) and related works on non-IID convergence. The constant $G$ quantifies cross-client heterogeneity. As Theorem 5.5 makes explicit later, $G$ appears in the bias term of the convergence bound. In particular, stronger heterogeneity does not break convergence but raises the asymptotic bias proportional to $G^2$.

## 5.1. Main Convergence Result

We first show that the admission window induces a bounded-staleness regime with high probability under mild sub-Gaussian queue-prediction errors and properly chosen safety buffer $\delta$. We emphasize that this concentration condition is imposed on the prediction error $e_k^{(r)} = q_k^{(r)} - \hat{q}_k^{(r)}$, not on the raw queue delay $q_k^{(r)}$, whose distribution may be heavy-tailed.

**Lemma 5.4** (Admission-Induced Bounded Staleness). *Fix $T_{sync} > 0$ and $\delta > 0$, and let $\gamma \geq 0$ be an analysis threshold. Assume the local training time respects the job-time budget, i.e., $h_k^{(r)} \leq J_k^{(r)}$ for all $k, r$.*

*Let $e_k^{(r)} := q_k^{(r)} - \hat{q}_k^{(r)}$ denote the queue-delay prediction error (Section 4.1.1), and let $\mathcal{F}_{r-1}$ denote the filtration up to round $r - 1$. Suppose that conditionally on the history up to round $r - 1$, $e_k^{(r)}$ is zero-mean and $\rho_k$-sub-Gaussian:*

$$\mathbb{E}\left[\exp\left(\lambda e_k^{(r)}\right) \mid \mathcal{F}_{r-1}\right] \leq \exp\left(\frac{\lambda^2 \rho_k^2}{2}\right) \quad \text{for all } \lambda \in \mathbb{R}. \quad (1)$$

*Consider the buffering rule in Section 3.3. If a job submitted at time $rT_{sync}$ completes by time $rT_{sync} + (1 + \gamma)T_{sync}$, then it must be incorporated within at most $\lceil 1 + \gamma \rceil$ subsequent server cutoffs, yielding a staleness bound.*

*Then, for any $\varepsilon \in (0, 1)$, if $\delta$ is chosen such that*

$$\gamma T_{sync} + \delta \geq \max_{k \in [K]} \sqrt{2\rho_k^2 \log\left(\frac{KR}{\varepsilon}\right)}, \quad (2)$$

*we have, with probability at least $1 - \varepsilon$,*

$$\tau_k^{(r)} \leq \tau_{\max} := \lceil 1 + \gamma \rceil \quad \text{for all } k, r. \quad (3)$$

We report empirical prediction error statistics, staleness distributions and violation frequencies that validate the practical tightness of this bound in Section 6.

We present our main convergence theorem for FEDQUEUE under non-convex objectives with bounded staleness and queue prediction error.

**Theorem 5.5** (Convergence of FEDQUEUE). *Under Assumptions 5.1–5.3, suppose the staleness is bounded by*

$\tau_k^{(r)} \leq \tau_{\max}$ *for all $k \in [K]$ and $r \in [R]$. Let $\eta_k^{(r)} = \eta_{base} E_{\min}^{(r)} / E_k^{(r)}$ with $\eta_{base} \leq \frac{1}{8LE_{\max}}$, where $E_{\max} = \max_{k,r} E_k^{(r)}$. Assume the staleness decay function satisfies $\phi(\tau) \geq \phi_{\min} > 0$ for all $\tau \leq \tau_{\max}$. Then, for the iterates generated by FEDQUEUE (Algorithm 1), there exist constants $C_0, C_1 > 0$ such that*

$$\frac{1}{R} \sum_{r=0}^{R-1} \mathbb{E}\left[\|\nabla F(w^{(r)})\|^2\right] \leq C_0 \frac{F(w^{(0)}) - F^*}{\eta_{base} E_{\min} R}$$
$$+ C_1 \eta_{base} E_{\max}\left(L^2 \tau_{\max}^2 + G^2 + \sigma^2\right).$$

*where $F^* = \inf_w F(w)$, $E_{\min} = \min_{k,r} E_k^{(r)}$, and $\sigma^2 = \max_k \sigma_k^2$.*

*Remark* 5.6 (Convergence). Theorem 5.5 matches the standard $\mathcal{O}(1/\sqrt{R})$ stationarity rate for non-convex FL when $\eta_{base} = \Theta(1/\sqrt{R})$. The bias scales with heterogeneity ($G^2$), stochasticity ($\sigma^2$), and the square of the staleness bound ($\tau_{\max}^2$), underscoring the importance of admission-induced staleness control. Our convergence bound deviates from the standard FL algorithm in three ways: 1) Our asynchronous approach allows updates from models up to $\tau_{\max}$ rounds stale, introducing the penalty term $L^2 \tau_{\max}^2$ in the bias. However, $\tau_{\max}$ is a controlled system parameter that does not grow with $R$. 2) We allow adaptive local steps, introducing $E_{\min}$ and $E_{\max}$. The optimization error scales with the slowest client while the bias scales with the fastest client. 3) We introduce adaptive learning rate that is computed based on $\eta_{base}$ which prevents clients with more local steps from dominating the updates. We demonstrate the convergence of FEDQUEUE empirically in Section 6.

*Remark* 5.7 (Queue Prediction Threshold). Lemma 5.4 provides a high-probability bound on staleness under the buffering rule by selecting an *analysis threshold* $\gamma$ and setting $\tau_{\max} = \lceil 1 + \gamma \rceil$. Importantly, $\gamma$ is not an algorithm parameter in FEDQUEUE; it only indexes the probabilistic arrival-time event used in the staleness bound.

The proofs of Lemma 5.4 and Theorem 5.5 are deferred to the Appendices A and B.

## 6. Experiments

FEDQUEUE is evaluated through two sets of experiments. First, Section 6.1 demonstrates superior time-to-quality and final model performance in a real-world cross-facility training on production HPC systems under unpredictable queue delays. Second, Section 6.2 uses controlled simulations to validate our convergence theory, quantify performance gains under varying queue variability and client arrivals while demonstrating resource efficiency improvements. Across these set of experiments, we use four baselines: FedAvg, FedAsync, FedBuff, and FedCompass. More details on the experimental setup are provided in Appendices D and E.

## 6.1. Large-Scale Cross-Facility Evaluation

**Experimental setup.** This experiment aims to measure efficiency of FEDQUEUE in real-world deployment. Therefore, four production HPC facilities participate as FL clients: Aurora and Polaris at Argonne Leadership Computing Facilities (ALCF), Perlmutter at the National Energy Research Scientific Computing Center (NERSC), and Frontier at Oak Ridge Leadership Computing Facility (OLCF). The FL server runs on a separate dedicated cluster. Each facility allocates two GPU nodes for client training and we use APPFL[1] (Li et al., 2025; Ryu et al., 2022) to implement all methods. We also use GLOBUS suite (Li et al., 2022; Zheng et al., 2024) to submit and monitor jobs across facilities.

**Model, dataset, and training.** We fine-tune a pretrained LLaMA2-7B model (Touvron et al., 2023) on SMolInstruct (Yu et al., 2024a), a curated chemistry instruction dataset partitioned across the four facilities by task categories. Each algorithm runs until a wall-clock budget of 17,000 seconds is exhausted and uses time-to-quality as a measurement to identify performance. To ensure fair comparison under realistic but comparable conditions, all algorithms were executed during roughly similar times of day, so that facilities experienced similar load patterns, while still being subject to the stochastic queue dynamics.

### 6.1.1. REAL-WORLD DEPLOYMENT EFFICIENCY

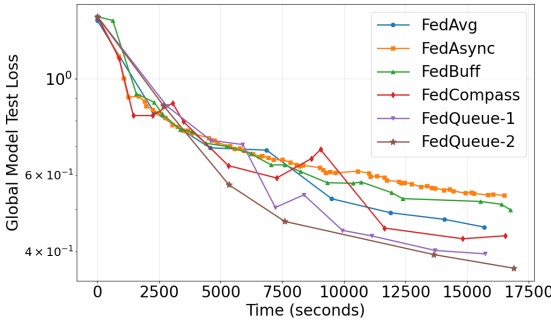

*Figure 2.* **Test loss of the federated global models versus wall-clock time across all algorithms.** There are two configurations of FEDQUEUE tested. FEDQUEUE-1 has $(T_{\text{sync}}, \delta) = (20\text{min}, 1\text{min})$ while FEDQUEUE-2 has $(T_{\text{sync}}, \delta) = (40\text{min}, 2\text{min})$. FEDQUEUE starts to achieve lower test loss once it builds accurate queue and compute estimates of the HPC systems and ultimately reaches the smallest test among the algorithms. Additional details on the setup is available in Appendix D.

**Time-to-threshold analysis.** Figure 2 shows global model test loss versus wall-clock time for all methods. To quantify deployment efficiency, we measure the time required for each algorithm to reach several target loss thresholds ($\ell \in \{1.0, 0.8, 0.6, 0.4\}$) and report the final loss values at the wall-clock budget. Table 1 summarizes these results with detailed facility-level loss trajectories provided in Ap-

---

[1]https://github.com/APPFL/APPFL

pendix D. As noted, FEDQUEUE achieves the best final loss values while being the only algorithm to attain loss below $0.4$ and reaches $0.6$ approximately $1.78\times$ faster than FedAvg and $1.36\times$ faster than the FedCompass. FedCompass degrades as queue conditions shift over the roughly 5-hour window even with impressive early performance.

*Table 1.* **Time-to-loss-threshold comparison.** Time required to reach target test loss thresholds for each algorithm. "—" indicates threshold not reached within 17,000 second budget. FedAsync converges quickly to early thresholds but plateaus at poor final loss. Only FEDQUEUE reaches loss value below $0.4$.

| Threshold | FedAvg | FedAsync | FedBuff | FedCompass | FEDQUEUE-1 | FEDQUEUE-2 |
|---|---|---|---|---|---|---|
| 1.0 | 2284 | **1245** | 1574 | 1454 | 2807 | 2680 |
| 0.8 | 4567 | **3038** | 3351 | 3490 | 4597 | 5331 |
| 0.6 | 9502 | 11079 | 9328 | 7270 | 7216 | 5331 |
| 0.4 | — | — | — | — | **15727** | 16881 |
| Final loss | 0.4549 | 0.5381 | 0.4987 | 0.4345 | 0.3947 | **0.3658** |

### 6.1.2. QUEUE DYNAMICS AND ADMISSION BEHAVIOR

**Observed queue patterns.** We use Globus Compute Endpoint to submit jobs to its native production scheduler (PBS or Slurm) with Table 2 detailing the queue conditions. Notably, queue times exhibit high variance across facilities with random delays, reflecting realistic heterogeneity in production HPC environments forcing static profiling approaches (e.g., FedCompass) to struggle in production settings.

*Table 2.* **Cross-facility systems and observed queue conditions.** Hardware configuration and queue behavior observed across four production HPC facilities. Facilities operate different schedulers with heterogeneous GPU counts. Queue times exhibit substantial cross-facility heterogeneity and high within-facility variance. Statistics computed over all jobs during the experimental period.

| Facility | Scheduler | GPUs | | Queue time (sec.) | |
|---|---|---|---|---|---|
| | | per node | per job | median | p90 |
| ALCF-Aurora | PBS | 12 | 24 | 1015.53 | 1638.38 |
| ALCF-Polaris | PBS | 4 | 8 | 471.15 | 1673.44 |
| OLCF-Frontier | Slurm | 8 | 16 | 741.14 | 1188.86 |
| NERSC-Perlmutter | Slurm | 4 | 8 | 673.68 | 1668.29 |

**Prediction error characterization.** We characterize empirically the queue-delay prediction error underlying the sub-Gaussian condition in Lemma 5.4, in addition to the evident cross-facility queue variability in Table 2. After excluding 1–2 rounds per facility corresponding to anomalous scheduler spikes, cleaned prediction errors have near-zero means and standard deviations of 39–114 sec—about 3–10% of $T_{\text{sync}}$ for FEDQUEUE-1 and 2–5% for FEDQUEUE-2—small relative to the synchronization horizon. These statistics are descriptive given the limited rounds per facility rather than a formal test of the tail condition; the assumption serves as a sufficient condition under which Lemma 5.4 yields bounded staleness, a consequence we validate directly in Table 3.

**Admission decisions and staleness.** Table 3 summarizes FEDQUEUE's admission behavior where FEDQUEUE admits 71.7% of submitted updates. It maintains bounded staleness with maximum observed staleness is one, consistent

with Lemma 5.4, suggesting that the anomalous scheduler spikes are handled gracefully by the buffering mechanism rather than causing staleness violations.

*Table 3.* **Combined admission and utilization statistics from the two runs of FEDQUEUE.** Max delay ratio for each HPC system is defined as $\max_r \left( \frac{t_k}{T_{\text{sync}}} \right)$.

| Facility | Jobs submitted | Admitted | Deferred | Max Delay↓ |
|---|---|---|---|---|
| ALCF-Aurora | 10 | 6 | 4 | 1.34 |
| ALCF-Polaris | 10 | 6 | 4 | 1.33 |
| OLCF-Frontier | 12 | 10 | 2 | 1.16 |
| NERSC-Perlmutter | 12 | 10 | 2 | **1.09** |

### 6.2. Controlled Synthetic Queue Experiments

To further understand the reasons behind impressive performance of FEDQUEUE, we run controlled experiments under synthetic queue dynamics. We defer extended analyses to Appendix E and focus on main results.

**Setup.** We simulate $K = 4$ clients with non-IID MNIST data partitions with a CNN and a cross-entropy loss. We measure test accuracy and fix the optimizer, batch size, evaluation cadence, and per-round resource requests across all baselines; differences are confined to orchestration (sync/async), budgeting, admission, and aggregation logic. Additional details are provided in Appendix E.1. We primarily report results with $K = 4$ in the main text; detailed scalability with $K = 8$ and $K = 12$ is deferred to Appendix E.3.

#### 6.2.1. CONVERGENCE UNDER QUEUE VARIABILITY

We study the impact of queue variability on convergence by sweeping the queue-noise parameter $\rho_k \in \{0.1, 0.5, 0.9\}$, while holding the model architecture, optimizer, batch size, and per-round resource requests fixed. Performance is evaluated using wall-clock *time-to-target accuracy* (95%), *maximum achieved accuracy*, and the *data movement ratio* required to reach 95% accuracy.

**FEDQUEUE consistently accelerates convergence under queue variability.** Across all queue regimes, FEDQUEUE achieves faster time-to-target accuracy than the other baselines with the gains becoming most pronounced under high queue variance. Specifically, when $\rho_k = 0.9$, referring to high queue variance, FEDQUEUE improves time-to-target by 37% relative to FedAvg, 35% relative to FedBuff, and 60% relative to FedAsync and outperforms FedCompass by approximately 39%. These improvements, indicated in Table 4 and Figure 3 cement that FEDQUEUE maintains rapid and stable convergence even when queue noise increases.

**Faster convergence is achieved with significantly improved resource efficiency.** Beyond wall-clock speedups, FEDQUEUE also exhibits superior communication efficiency and local resource utilization. Analysis of the model move-

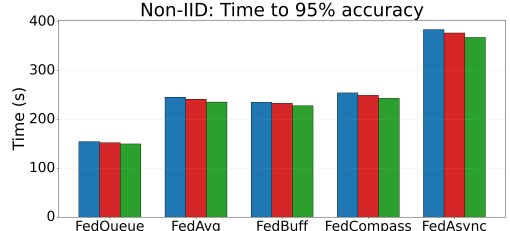

*Figure 3.* **Time-to-quality.** Validation accuracy vs. elapsed time to reach 95% accuracy under increasing queue variance $\rho_k$ (■ $\rho$=0.9, ■ $\rho$=0.5, ■ $\rho$=0.1.)

ment ratio $D_r$ and total local steps $\#E_k$ in Table 4 shows that FEDQUEUE saves model transfers by up to 67%, and approximately $2.1\times$ reduction in local resource usage as compared to the baselines. These speedups persist as the number of clients grows: under the same high-variance non-IID regime, FEDQUEUE achieves up to $1.7\times$ speedup and over $3\times$ reduction in local resource usage (Appendix E.3).

*Table 4.* **Synthetic summary.** Max accuracy, time-to-target, and model movement ratio ($D_r$), and total local steps ($\#E_k$) to 95% accuracy under a fixed non-IID partition and high queue variability ($\rho = 0.9$).

| Method | Max-A↑ | Time-to-$A^\star$ ↓ (s) | $D_r$ ↓ | $\#E_k$ ↓ |
|---|---|---|---|---|
| FEDQUEUE | $96.62 \pm 0.18$ | $\mathbf{154.72 \pm 5.56}$ | 1.00 | **5893** |
| FedAvg | $95.63 \pm 0.69$ | $245.60 \pm 4.29$ | 1.51 | 11520 |
| FedAsync | $95.75 \pm 0.94$ | $384.07 \pm 8.18$ | 1.67 | 12400 |
| FedBuff | $97.45 \pm 0.16$ | $235.10 \pm 4.27$ | 1.11 | 7440 |
| FedCompass | $\mathbf{97.21 \pm 0.35}$ | $254.44 \pm 7.79$ | 1.15 | 8449 |

#### 6.2.2. SENSITIVITY TO ARRIVAL VARIABILITY.

**The safety buffer $\delta$ controls a fundamental trade-off: staleness vs. convergence speed.** In our budgeting rule, $\delta$ provides slack against queue-delay uncertainty, making the per-round job-time budget more conservative. Larger $\delta$ increases the fraction of clients that finish and arrive before the cutoff $T_{\text{sync}}$, thereby concentrating staleness and improving stability (as in Lemma 5.4), but it also reduces the effective amount of local progress per time, which can slow the time to $A^*$. Figure 4 demonstrates this effect: larger $\delta$ (green histogram) leads to a histogram whose tail does not cross the $T_{\text{sync}}$ line, but achieves slower time to $A^*$ (right panel). Conversely, smaller $\delta$ (blue histogram) crosses $T_{\text{sync}}$ more frequently, implying more buffered (stale) updates, but it reaches $A^*$ faster due to more aggressive local work.

Queue variability $\rho$ and safety buffer $\delta$ are typically not in the designer's control, as they depend on HPC system characteristics and scheduler behavior. Our theory and controlled experiments show that FEDQUEUE is robust to variation in queue dynamics and client arrival patterns. Appendix E.2 provides detailed analysis of how sweeping $\rho$ and $\gamma$ affects client arrival probability beyond $T_{\text{sync}}$, time-to-quality, and maximum delay.

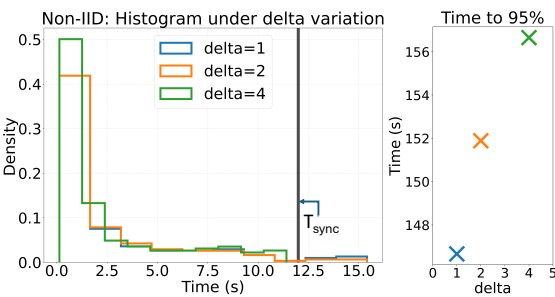

*Figure 4.* **Impact of admission buffer.** (Left) Effect of scaling the buffer $\delta$ on histogram of arrival times and (Right) corresponding time to quality (95%). This test exposes the trade-off between better convergence value and convergence time in the presence of client arrival variability. Note that the histograms of $\delta = 1, 2$ are almost identical and overlapped each other.

### 6.2.3. ABLATION STUDIES

There are three main components in FEDQUEUE: EWMA, staleness decay and inverse LR. We ablate these components to understand their impact on performance. We consider: (i) **w/o EWMA**: replace queue prediction by a static estimate $\hat{q}_k^{(r)} \equiv \mu_k$. (ii) **w/o staleness decay**: set $\phi(\tau) \equiv 1$ in aggregation weights, and (iii) **w/o inverse LR scaling**: use a fixed learning rate per facility independent of $E_k^{(r)}$. These results are presented in Table 5 with 85% target accuracy.

*Table 5.* **Ablation results.** Final Accuracy, time-to-target accuracy (85%), empirical probability of arrival beyond the cutoff ($\mathbb{P}$), Normalized expectation of delay ($\hat{\mathbb{E}}_d$), Normalized maximum delay ($R_d$) for different FEDQUEUE variants.

| Method variant | Final $A\%$ ↑ | Time-to-85% (s) ↓ | $\mathbb{P}$ ↓ | $\hat{\mathbb{E}}_d$ ↓ | $R_d$ ↓ |
|---|---|---|---|---|---|
| FEDQUEUE (Baseline) | 96.62 | 26.68 | 0.015 | 1.13 | 1.17 |
| w/o inverse LR | **90.89** | 28.13 | 0.020 | 1.15 | 1.16 |
| w/o EWMA | 96.01 | **116.59** | 0.035 | 1.32 | 1.38 |
| w/o staleness decay | **87.18** | 34.61 | 0.015 | 1.14 | 1.17 |

**Queue prediction matters.** Removing EWMA degrades performance (time to 85%) by 330% and normalized expected delay ($\hat{\mathbb{E}}_d$) by 17%, despite comparable final accuracy. Therefore, EWMA is essential in the presence of non-stationary scheduler dynamics.

**Staleness weighting is stabilizing.** Disabling staleness decay ($\phi(\tau) \equiv 1$) reduces final accuracy by approximately 10% and increases time-to-target by approximately 30%, a behavior that aligns with Theorem 5.5. Notably without decay, stale updates contribute disproportionately to updates increasing optimization variance.

**Inverse LR scaling prevents domination.** Removing inverse learning-rate scaling leads to an approximately 6% drop in final accuracy and an approximately 5% increase in time-to-target.

Notably staleness significantly impacts accuracy while EWMA significantly impacts solution time. It is expected

that, this trade-off can be decided by choosing the values of $\alpha$ and $\gamma$ depending on the requirement of the practitioner. However, FEDQUEUE allows this flexibility.

### 6.3. Additional Dataset

We additionally evaluate FEDQUEUE on CIFAR-100 with a ResNet-18 backbone, holding the queue regimes and configurations fixed across methods. We measure time and local steps ($\#E_k$), alongside maximum accuracy, results are presented in Table 6. Under the IID partition, FEDQUEUE achieves performance comparable to FedAvg, matching it on both time and local steps to threshold. FedBuff and FedCompass attain higher maximum accuracy but require roughly $2\times$ the local steps to do so. Under the non-IID partition, FEDQUEUE is the only method to reach $45\%$ accuracy within the wall-clock budget, and attains the highest maximum accuracy overall, mirroring the pattern observed on MNIST.

*Table 6.* CIFAR-100 with ResNet-18. Max accuracy, time and total local steps under IID and non-IID partitions. "–" denotes the threshold was never reached.

| | IID | | | Non-IID | | |
|---|---|---|---|---|---|---|
| Method | Max-A ↑ | Time ↓ | $\#E_k$ ↓ | Max-A ↑ | Time ↓ | $\#E_k$ ↓ |
| FedAvg | 50.6 | **315.2** | **14208** | 42.6 | – | – |
| FedAsync | 52.8 | 698.2 | 40300 | 41.6 | – | – |
| FedBuff | **56.4** | 465.0 | 26660 | 44.6 | – | – |
| FedCompass | 55.6 | 1368.0 | 25774 | 44.7 | – | – |
| FEDQUEUE | 51.7 | 384.0 | 14422 | **46.9** | **465.1** | **18332** |

## 7. Conclusion and Limitations

We introduced FEDQUEUE, a queue-aware federated learning protocol that addresses the challenge of unpredictable scheduler admission delays in cross-facility training. FEDQUEUE achieves provably bounded staleness and maintains convergence guarantees under stochastic queue dynamics. Experiments demonstrate substantial improvements over existing approaches in both real-world cross-facility deployment and controlled simulations.

Although our buffering mechanism handles late arrivals gracefully, we do not model job failures. A failed job is indistinguishable from a late arrival and results in a lost update from the server. Most HPC schedulers support automatic requeue for preempted jobs, but explicit failure detection and resubmission of permanent failures are natural extensions. Our EWMA-based predictor could be substituted by approaches that are more responsive to abrupt step-changes in queue characteristics; because our convergence analysis is prediction-agnostic, ML-based predictors or schemes that query scheduler state directly can be substituted. Finally, our cross-facility setting naturally involves a small number of HPC clients and complementary directions such as data-quality-aware aggregation are left to future work.

## Acknowledgment

This work was supported by the U.S. Department of Energy, Office of Science, Advanced Scientific Computing Research, under Contract DE-AC02-06CH11357. An award of computer time was provided by the ASCR Leadership Computing Challenge (ALCC) program. This research used resources of the Argonne Leadership Computing Facility, which is a U.S. Department of Energy Office of Science User Facility operated under contract DE-AC02-06CH11357. This research used resources of the Oak Ridge Leadership Computing Facility at the Oak Ridge National Laboratory, which is supported by the Office of Science of the U.S. Department of Energy under Contract No. DE-AC05-00OR22725. This research used resources of the National Energy Research Scientific Computing Center (NERSC), a Department of Energy User Facility using NERSC award ALCC-ERCAP0038201. We gratefully acknowledge the computing resources provided on Improv, a high-performance computing cluster operated by the Laboratory Computing Resource Center at Argonne National Laboratory.

## Impact Statement

This paper presents work whose goal is to advance the field of Machine Learning. There are many potential societal consequences of our work, none which we feel must be specifically highlighted here.

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

# A. Proof of Lemma 5.4

Fix any facility $k \in [K]$ and round $r \in \{0, \ldots, R-1\}$. By definition of $J_k^{(r)}$ and assuming that local training time respects the job-time budget, i.e., $h_k^{(r)} \leq J_k^{(r)}$ for all $k, r$,

$$h_k^{(r)} \leq J_k^{(r)} = T_{\text{sync}} - \hat{q}_k^{(r)} - \delta.$$

Hence the total completion time of the job submitted at time $rT_{\text{sync}}$ satisfies

$$a_k^{(r)} = rT_{\text{sync}} + q_k^{(r)} + h_k^{(r)} \leq rT_{\text{sync}} + q_k^{(r)} + T_{\text{sync}} - \hat{q}_k^{(r)} - \delta = rT_{\text{sync}} + T_{\text{sync}} + e_k^{(r)} - \delta,$$

where $e_k^{(r)} = q_k^{(r)} - \hat{q}_k^{(r)}$. Therefore, if the prediction error satisfies

$$e_k^{(r)} \leq \gamma T_{\text{sync}} + \delta, \tag{4}$$

then the completion time bound above implies

$$a_k^{(r)} \leq rT_{\text{sync}} + T_{\text{sync}} + (\gamma T_{\text{sync}} + \delta) - \delta = rT_{\text{sync}} + (1 + \gamma)T_{\text{sync}}.$$

Under the buffering rule in Section 3.3 (aggregation at cutoffs $(j+1)T_{\text{sync}}$), any update that arrives by time $rT_{\text{sync}} + (1 + \gamma)T_{\text{sync}}$ is incorporated by at most $\lceil 1 + \gamma \rceil$ subsequent cutoffs, hence its staleness is at most $\tau_{\max} := \lceil 1 + \gamma \rceil$.

Using the sub-Gaussian assumption (1) and standard tail bounds, we obtain for any $u > 0$

$$\mathbb{P}\left(e_k^{(r)} > u \mid \mathcal{F}_{r-1}\right) \leq \exp\left(-\frac{u^2}{2\rho_k^2}\right).$$

Set $u = \gamma T_{\text{sync}} + \delta$. Then

$$\mathbb{P}\left(e_k^{(r)} > \gamma T_{\text{sync}} + \delta \mid \mathcal{F}_{r-1}\right) \leq \exp\left(-\frac{(\gamma T_{\text{sync}} + \delta)^2}{2\rho_k^2}\right).$$

By the choice of $\delta$ in (2), we have

$$\exp\left(-\frac{(\gamma T_{\text{sync}} + \delta)^2}{2\rho_k^2}\right) \leq \frac{\varepsilon}{KR} \quad \text{for all } k \in [K].$$

Thus, for each fixed $(k, r)$,

$$\mathbb{P}\left(e_k^{(r)} > \gamma T_{\text{sync}} + \delta\right) \leq \frac{\varepsilon}{KR}.$$

We now apply a union bound over all $k \in [K]$ and $r \in \{0, \ldots, R-1\}$:

$$\mathbb{P}\left(\exists k, r \text{ such that } e_k^{(r)} > \gamma T_{\text{sync}} + \delta\right) \leq KR \cdot \frac{\varepsilon}{KR} = \varepsilon.$$

Hence, with probability at least $1 - \varepsilon$, we have

$$e_k^{(r)} \leq \gamma T_{\text{sync}} + \delta \quad \text{for all } k, r.$$

Next we relate completion time to staleness under the buffering rule (Section 3.3). If a job submitted at time $rT_{\text{sync}}$ completes by $rT_{\text{sync}} + (1 + \gamma)T_{\text{sync}}$, then at most $\lceil 1 + \gamma \rceil$ end-of-round cutoffs elapse between submission and completion. Therefore, the update is incorporated within staleness at most $\tau_{\max} := \lceil 1 + \gamma \rceil$.

Finally, we absorb the factor $\varepsilon/2$ into $\varepsilon$ by redefining the target failure probability (or by tightening the constant in (2)). Thus, for any prescribed $\varepsilon \in (0, 1)$, by choosing $\delta$ according to (2) (with a slightly larger constant if needed), we obtain

$$\mathbb{P}\left(\tau_k^{(r)} \leq \tau_{\max} \text{ for all } k, r\right) \geq 1 - \varepsilon,$$

which proves (3). $\square$

# B. Proof of Theorem 5.5

We provide a convergence proof for FEDQUEUE under Assumptions 5.1–5.3 and bounded staleness $\tau_k^{(r)} \leq \tau_{\max}$.

Recall the notation: $w^{(r)}$ is the global model at round $r$, $R_k^{(r)}$ is the round index of the global model used by client $k$ to compute its update incorporated at round $r$, $\tau_k^{(r)} = r - R_k^{(r)}$ is the staleness, and $d_k^{(r)} := w^{(r)} - w^{(R_k^{(r)})}$ is the model drift for client $k$ at round $r$.

The local update from client $k$ at round $r$ is $\Delta_k^{(r)} = w_k^{(r, E_k^{(r)})} - w^{(R_k^{(r)})}$, and the global update is

$$\Delta^{(r)} := w^{(r+1)} - w^{(r)} = \frac{1}{S^{(r)}} \sum_{k \in A^{(r)}} p_k \phi(\tau_k^{(r)}) \Delta_k^{(r)}, \quad S^{(r)} := \sum_{k \in A^{(r)}} p_k \phi(\tau_k^{(r)}).$$

We denote $E_{\min} := \min_{k,r} E_k^{(r)}$.

## B.1. One-Step Descent Inequality

**Lemma B.1** (Descent under staleness). *Under Assumption 5.1, for any round $r$, we have:*

$$F(w^{(r+1)}) \leq F(w^{(r)}) + \frac{1}{S^{(r)}} \sum_{k \in \mathcal{A}^{(r)}} p_k \phi(\tau_k^{(r)}) \langle \nabla F(w^{(R_k^{(r)})}), \Delta_k^{(r)} \rangle$$

$$+ \frac{1}{S^{(r)}} \sum_{k \in \mathcal{A}^{(r)}} p_k \phi(\tau_k^{(r)}) \left( \frac{L^2}{2} \|d_k^{(r)}\|^2 + \frac{1}{2} \|\Delta_k^{(r)}\|^2 \right) + \frac{L}{2} \|\Delta^{(r)}\|^2. \tag{5}$$

*Proof.* By Assumption 5.1 (L-smoothness),

$$F(w^{(r+1)}) \leq F(w^{(r)}) + \langle \nabla F(w^{(r)}), \Delta^{(r)} \rangle + \frac{L}{2} \|\Delta^{(r)}\|^2.$$

Using the aggregation rule,

$$\langle \nabla F(w^{(r)}), \Delta^{(r)} \rangle = \left\langle \nabla F(w^{(r)}), \frac{1}{S^{(r)}} \sum_{k \in \mathcal{A}^{(r)}} p_k \phi(\tau_k^{(r)}) \Delta_k^{(r)} \right\rangle = \frac{1}{S^{(r)}} \sum_{k \in \mathcal{A}^{(r)}} p_k \phi(\tau_k^{(r)}) \langle \nabla F(w^{(r)}), \Delta_k^{(r)} \rangle.$$

For each $k \in \mathcal{A}^{(r)}$,

$$\langle \nabla F(w^{(r)}), \Delta_k^{(r)} \rangle = \langle \nabla F(w^{(R_k^{(r)})}), \Delta_k^{(r)} \rangle + \langle \nabla F(w^{(r)}) - \nabla F(w^{(R_k^{(r)})}), \Delta_k^{(r)} \rangle.$$

By Lipschitz continuity and Young's inequality,

$$|\langle \nabla F(w^{(r)}) - \nabla F(w^{(R_k^{(r)})}), \Delta_k^{(r)} \rangle| \leq \|\nabla F(w^{(r)}) - \nabla F(w^{(R_k^{(r)})})\| \|\Delta_k^{(r)}\|$$

$$\leq L \|w^{(r)} - w^{(R_k^{(r)})}\| \|\Delta_k^{(r)}\| = L \|d_k^{(r)}\| \|\Delta_k^{(r)}\|$$

$$\leq \frac{L^2}{2} \|d_k^{(r)}\|^2 + \frac{1}{2} \|\Delta_k^{(r)}\|^2.$$

Thus,

$$\langle \nabla F(w^{(r)}), \Delta_k^{(r)} \rangle \leq \langle \nabla F(w^{(R_k^{(r)})}), \Delta_k^{(r)} \rangle + \frac{L^2}{2} \|d_k^{(r)}\|^2 + \frac{1}{2} \|\Delta_k^{(r)}\|^2.$$

Substituting back completes the proof. $\square$

## B.2. Staleness Drift Accumulation

**Lemma B.2** (Drift bound). *If $\tau_k^{(r)} \leq \tau_{\max}$ for all $k, r$, then*

$$\|d_k^{(r)}\|^2 \leq \tau_{\max} \sum_{t=R_k^{(r)}}^{r-1} \|\Delta^{(t)}\|^2 \leq \tau_{\max} \sum_{t=0}^{r-1} \|\Delta^{(t)}\|^2.$$

*Proof.* The drift is the cumulative change in the global model over $\tau_k^{(r)}$ rounds:

$$d_k^{(r)} = w^{(r)} - w^{(R_k^{(r)})} = \sum_{t=R_k^{(r)}}^{r-1} (w^{(t+1)} - w^{(t)}) = \sum_{t=R_k^{(r)}}^{r-1} \Delta^{(t)}.$$

By the triangle inequality:

$$\|d_k^{(r)}\| = \left\| \sum_{t=R_k^{(r)}}^{r-1} \Delta^{(t)} \right\| \leq \sum_{t=R_k^{(r)}}^{r-1} \|\Delta^{(t)}\|.$$

By Cauchy-Schwarz:

$$\left( \sum_{t=R_k^{(r)}}^{r-1} \|\Delta^{(t)}\| \right)^2 \leq (r - R_k^{(r)}) \sum_{t=R_k^{(r)}}^{r-1} \|\Delta^{(t)}\|^2 = \tau_k^{(r)} \sum_{t=R_k^{(r)}}^{r-1} \|\Delta^{(t)}\|^2.$$

Since $\tau_k^{(r)} \leq \tau_{\max}$, the result follows. $\qquad\square$

### B.3. Local Update Variance and Bias

**Lemma B.3** (Local update decomposition). *Under Assumptions 5.1–5.3, for each facility $k$ performing $E_k^{(r)}$ local SGD steps starting from $w^{(R_k^{(r)})}$ with learning rate $\eta_k^{(r)} \leq \frac{1}{4LE_k^{(r)}}$, there exist absolute constants $c_1, c_2, c_3 > 0$ (independent of $k, r$) such that*

$$\mathbb{E}[\langle \nabla F(w^{(R_k^{(r)})}), \Delta_k^{(r)} \rangle \mid w^{(R_k^{(r)})}] \leq -c_1 \eta_k^{(r)} E_k^{(r)} \|\nabla F(w^{(R_k^{(r)})})\|^2 + c_2 (\eta_k^{(r)})^2 (E_k^{(r)})^2 (G^2 + \sigma_k^2), \tag{6}$$

*and*

$$\mathbb{E}[\|\Delta_k^{(r)}\|^2 \mid w^{(R_k^{(r)})}] \leq c_3 (\eta_k^{(r)})^2 (E_k^{(r)})^2 \left( \|\nabla F(w_k^{(R_k^{(r)})})\|^2 + G^2 + \sigma_k^2 \right). \tag{7}$$

*Proof.* Recall that facility $k$ performs local SGD starting from $w_k^{(r,0)} = w^{(R_k^{(r)})}$:

$$w_k^{(r,e+1)} = w_k^{(r,e)} - \eta_k^{(r)} g_k^{(r,e)},$$

where $g_k^{(r,e)}$ is a stochastic gradient sampled from $\mathcal{D}_k$. The local update is:

$$\Delta_k^{(r)} = w_k^{(r,E_k^{(r)})} - w^{(R_k^{(r)})} = -\eta_k^{(r)} \sum_{e=0}^{E_k^{(r)}-1} g_k^{(r,e)}.$$

Taking expectation:

$$\mathbb{E}[\Delta_k^{(r)} \mid w^{(R_k^{(r)})}] = -\eta_k^{(r)} \sum_{e=0}^{E_k^{(r)}-1} \mathbb{E}[g_k^{(r,e)} \mid w^{(R_k^{(r)})}] = -\eta_k^{(r)} \sum_{e=0}^{E_k^{(r)}-1} \nabla F_k(w_k^{(r,e)}).$$

We want to bound:

$$\mathbb{E}[\langle \nabla F(w^{(R_k^{(r)})}), \Delta_k^{(r)} \rangle \mid w^{(R_k^{(r)})}] = \left\langle \nabla F(w^{(R_k^{(r)})}), \mathbb{E}[\Delta_k^{(r)} \mid w^{(R_k^{(r)})}] \right\rangle$$

$$= -\eta_k^{(r)} \sum_{e=0}^{E_k^{(r)}-1} \mathbb{E}\left[ \langle \nabla F(w^{(R_k^{(r)})}), \nabla F_k(w_k^{(r,e)}) \rangle \mid w^{(R_k^{(r)})} \right]. \tag{8}$$

For each term in the sum, we have

$$\langle \nabla F(w^{(R_k^{(r)})}), \nabla F_k(w_k^{(r,e)}) \rangle \geq \frac{1}{2} \|\nabla F(w^{(R_k^{(r)})})\|^2 - \frac{1}{2} \|\nabla F(w^{(R_k^{(r)})}) - \nabla F_k(w_k^{(r,e)})\|^2$$

$$\geq \frac{1}{2} \|\nabla F(w^{(R_k^{(r)})})\|^2$$

$$- \frac{1}{2} \left( 2\|\nabla F(w^{(R_k^{(r)})}) - \nabla F_k(w_k^{(R_k^{(r)})})\|^2 + 2\|\nabla F_k(w^{(R_k^{(r)})}) - \nabla F_k(w_k^{(r,e)})\|^2 \right)$$

$$\geq \frac{1}{2} \|\nabla F(w^{(R_k^{(r)})})\|^2 - G^2 - L^2 \|w^{(R_k^{(r)})} - w_k^{(r,e)}\|^2$$

From the local SGD update:

$$\|w^{(R_k^{(r)})} - w_k^{(r,e)}\|^2 = (\eta_k^{(r)})^2 \left\| \sum_{j=0}^{e-1} g_k^{(r,j)} \right\|^2 \leq (\eta_k^{(r)})^2 e \sum_{j=0}^{e-1} \|g_k^{(r,j)}\|^2,$$

where the inequality uses Cauchy-Schwarz. Taking expectations and using the variance bound:

$$\mathbb{E}\left[ \|g_k^{(r,j)}\|^2 \mid w_k^{(R_k^{(r)})} \right] \leq 2\mathbb{E}\left[ \|\nabla F_k(w_k^{(r,j)})\|^2 \mid w_k^{(R_k^{(r)})} \right] + 2\sigma_k^2.$$

Now we relate $\|\nabla F_k(w_k^{(r,j)})\|^2$ to $\nabla F(w_k^{(r,j)})$ using Assumption 5.3,

$$\|\nabla F_k(w_k^{(r,j)})\| \leq \|\nabla F(w_k^{(r,j)})\| + \|\nabla F_k(w_k^{(r,j)}) - \nabla F(w_k^{(r,j)})\| \leq \|\nabla F(w_k^{(r,j)})\| + G,$$

so $\|\nabla F_k(w_k^{(r,j)})\|^2 \leq \|\nabla F(w_k^{(r,j)})\|^2 + G^2$. By Assumption 5.1,

$$\|\nabla F(w_k^{(r,j)})\|^2 \leq 2\|\nabla F(w_k^{(R_k^{(r)})})\|^2 + 2L^2 \|w_k^{(r,j)} - w_k^{(R_k^{(r)})}\|^2.$$

Combining these gives

$$\mathbb{E}\left[ \|g_k^{(r,j)}\|^2 \mid w_k^{(R_k^{(r)})} \right] \leq C_0 \left( \|\nabla F(w_k^{(R_k^{(r)})})\|^2 + G^2 + \sigma_k^2 + L^2 \mathbb{E}\left[ \|w_k^{(r,j)} - w_k^{(R_k^{(r)})}\|^2 \mid w_k^{(R_k^{(r)})} \right] \right),$$

for a constant $C_0$. Plugging into the drift bound,

$$\mathbb{E}\left[ \|w^{(R_k^{(r)})} - w_k^{(r,e)}\|^2 \mid w^{(R_k^{(r)})} \right]$$

$$\leq C_0 (\eta_k^{(r)})^2 e \sum_{j=0}^{e-1} \left( \|\nabla F(w_k^{(R_k^{(r)})})\|^2 + G^2 + \sigma_k^2 + L^2 \mathbb{E}\left[ \|w_k^{(r,j)} - w_k^{(R_k^{(r)})}\|^2 \mid w_k^{(R_k^{(r)})} \right] \right)$$

Using the small step-size condition $\eta \leq 1/(4LE)$, one can show by induction on $e$ that the $L^2 \mathbb{E}\|w_j - w_0\|^2$ part does not explode and can be absorbed into the constant (this is a standard argument in non-convex SGD analyses (e.g., Nguyen et al. (2022)). Concretely, there exists a constant $C_1 > 0$ such that

$$\mathbb{E}\left[ \|w^{(R_k^{(r)})} - w_k^{(r,e)}\|^2 \mid w^{(R_k^{(r)})} \right] \leq C_1 (\eta_k^{(r)})^2 e^2 \left( \|\nabla F(w_k^{(R_k^{(r)})})\|^2 + G^2 + \sigma_k^2 \right)$$

Summing over $e$ gives

$$\sum_{e=0}^{E_k^{(r)}-1} \mathbb{E}\left[ \|w^{(R_k^{(r)})} - w_k^{(r,e)}\|^2 \mid w^{(R_k^{(r)})} \right] \leq C_1 (\eta_k^{(r)})^2 \left( \|\nabla F(w_k^{(R_k^{(r)})})\|^2 + G^2 + \sigma_k^2 \right) \sum_{e=0}^{E_k^{(r)}-1} e^2$$

$$\leq C_2 (\eta_k^{(r)})^2 (E_k^{(r)})^3 \left( \|\nabla F(w_k^{(R_k^{(r)})})\|^2 + G^2 + \sigma_k^2 \right),$$

where the last inequality uses $\sum_{e=0}^{E-1} e^2 \leq E^3/3$. Substituting the bounds to (8),

$$
\begin{aligned}
\mathbb{E}[\langle \nabla F(w^{(R_k^{(r)})}), \Delta_k^{(r)}\rangle \mid w^{(R_k^{(r)})}] \leq{} & -\frac{\eta_k^{(r)} E_k^{(r)}}{2}\|\nabla F(w^{(R_k^{(r)})})\|^2 + \eta_k^{(r)} E_k^{(r)} G^2 \\
& + \eta_k^{(r)} L^2 \sum_{e=0}^{E_k^{(r)}-1} \mathbb{E}\left[\|w^{(R_k^{(r)})} - w_k^{(r,e)}\|^2 \mid w^{(R_k^{(r)})}\right] \\
\leq{} & -\frac{\eta_k^{(r)} E_k^{(r)}}{2}\|\nabla F(w^{(R_k^{(r)})})\|^2 + \eta_k^{(r)} E_k^{(r)} G^2 \\
& + C_2 L^2 (\eta_k^{(r)})^3 (E_k^{(r)})^3 \left(\|\nabla F(w_k^{(R_k^{(r)})})\|^2 + G^2 + \sigma_k^2\right).
\end{aligned}
$$

Under the step-size constraint $\eta_k^{(r)} \leq \frac{1}{4LE_k^{(r)}}$, the last term can be bounded as $C_2 L^2 (\eta_k^{(r)})^3 (E_k^{(r)})^3 \leq \frac{C_2}{64L}$, so for some constant $c_1 \in (0, 0.5)$ and $c_2 > 0$, we obtain the result (6).

To derive the variance bound:

$$
\mathbb{E}\left[\|\Delta_k^{(r)}\|^2 \mid w^{(R_k^{(r)})}\right] = (\eta_k^{(r)})^2 \mathbb{E}\left[\left\|\sum_{e=0}^{E_k^{(r)}-1} g_k^{(r,e)}\right\|^2 \mid w^{(R_k^{(r)})}\right] \leq (\eta_k^{(r)})^2 E_k^{(r)} \sum_{e=0}^{E_k^{(r)}-1} \mathbb{E}\left[\left\|g_k^{(r,e)}\right\|^2 \mid w^{(R_k^{(r)})}\right],
$$

where the inequality holds by Cauchy-Schwarz. Similarly,

$$
\begin{aligned}
\mathbb{E}\left[\|\Delta_k^{(r)}\|^2 \mid w^{(R_k^{(r)})}\right] &\leq (\eta_k^{(r)})^2 E_k^{(r)} \sum_{e=0}^{E_k^{(r)}-1} \mathbb{E}\left[\left\|g_k^{(r,e)}\right\|^2 \mid w^{(R_k^{(r)})}\right] \\
&\leq c_3 (\eta_k^{(r)})^2 (E_k^{(r)})^2 \left(\|\nabla F(w_k^{(R_k^{(r)})})\|^2 + G^2 + \sigma_k^2\right).
\end{aligned}
$$

This completes the proof. $\qquad\square$

### B.4. Aggregate Update Norm

**Lemma B.4** (Aggregate norm and weights). *For any round $r$,*

$$
\|\Delta^{(r)}\|^2 \leq \frac{1}{S^{(r)}} \sum_{k \in \mathcal{A}^{(r)}} p_k \phi(\tau_k^{(r)})\|\Delta_k^{(r)}\|^2. \tag{9}
$$

*Proof.* The global update is bounded as follows:

$$
\begin{aligned}
\|\Delta^{(r)}\|^2 &= \left\|\frac{1}{S^{(r)}} \sum_{k \in \mathcal{A}^{(r)}} p_k \phi(\tau_k^{(r)})\Delta_k^{(r)}\right\|^2 \\
&\leq \frac{1}{(S^{(r)})^2}\left(\sum_{k \in \mathcal{A}^{(r)}} p_k \phi(\tau_k^{(r)})\right)\left(\sum_{k \in \mathcal{A}^{(r)}} p_k \phi(\tau_k^{(r)})\|\Delta_k^{(r)}\|^2\right) = \frac{1}{S^{(r)}} \sum_{k \in \mathcal{A}^{(r)}} p_k \phi(\tau_k^{(r)})\|\Delta_k^{(r)}\|^2,
\end{aligned}
$$

where the inequality holds due to Cauchy-Schwarz. $\qquad\square$

## B.5. Telescoping Sum and Final Convergence Rate

We now combine Lemmas B.1–B.4 to prove Theorem 5.5. We take expectations in Lemma B.1 (conditional on the history up to round $r$):

$$
\mathbb{E}[F(w^{(r+1)})] \leq \mathbb{E}[F(w^{(r)})]
$$

$$
+ \frac{1}{S^{(r)}} \sum_{k \in \mathcal{A}^{(r)}} p_k \phi(\tau_k^{(r)}) \mathbb{E}[\langle \nabla F(w^{(R_k^{(r)})}), \Delta_k^{(r)} \rangle] \qquad \text{(Progress)}
$$

$$
+ \frac{1}{S^{(r)}} \sum_{k \in \mathcal{A}^{(r)}} p_k \phi(\tau_k^{(r)}) \frac{L^2}{2} \mathbb{E}[\|d_k^{(r)}\|^2] \qquad \text{(Staleness)}
$$

$$
+ \frac{1}{S^{(r)}} \sum_{k \in \mathcal{A}^{(r)}} p_k \phi(\tau_k^{(r)}) \frac{1}{2} \mathbb{E}[\|\Delta_k^{(r)}\|^2] \qquad \text{(Local Variance)}
$$

$$
+ \frac{L}{2} \mathbb{E}[\|\Delta^{(r)}\|^2]. \qquad \text{(Global Variance)}
$$

By Lemma B.3,

$$
\mathbb{E}[\langle \nabla F(w^{(R_k^{(r)})}), \Delta_k^{(r)} \mid w^{(R_k^{(r)})} \rangle] \leq -c_1 \eta_k^{(r)} E_k^{(r)} \|\nabla F(w^{(R_k^{(r)})})\|^2 + c_2 (\eta_k^{(r)})^2 (E_k^{(r)})^2 (G^2 + \sigma_k^2)
$$

$$
\leq -c_1 \eta_{\text{base}} E_{\min} \|\nabla F(w^{(R_k^{(r)})})\|^2 + c_2 \eta_{\text{base}}^2 E_{\min}^2 (G^2 + \sigma^2)
$$

Substituting this into the Progress term gives:

$$
\text{(Progress)} \leq -\frac{c_1 \eta_{\text{base}} E_{\min}}{S^{(r)}} \sum_{k \in \mathcal{A}^{(r)}} p_k \phi(\tau_k^{(r)}) \mathbb{E}[\|\nabla F(w^{(R_k^{(r)})})\|^2] + c_2 \eta_{\text{base}}^2 E_{\min}^2 (G^2 + \sigma^2)
$$

To relate $\|\nabla F(w^{(R_k^{(r)})})\|^2$ to $\|\nabla F(w^{(r)})\|^2$, we use Assumption 5.1:

$$
\|\nabla F(w^{(R_k^{(r)})}) - \nabla F(w^{(r)})\| \leq L \|d_k^{(r)}\|,
$$

which implies

$$
\|\nabla F(w^{(R_k^{(r)})})\|^2 \geq \frac{1}{2} \|\nabla F(w^{(r)})\|^2 - L^2 \|d_k^{(r)}\|^2.
$$

Combining these, we get:

$$
\text{(Progress)} \leq -\frac{c_1 \eta_{\text{base}} E_{\min}}{S^{(r)}} \sum_{k \in \mathcal{A}^{(r)}} p_k \phi(\tau_k^{(r)}) \left( \frac{1}{2} \mathbb{E}[\|\nabla F(w^{(r)})\|^2] - L^2 \mathbb{E}[\|d_k^{(r)}\|^2] \right) + c_2 \eta_{\text{base}} E_{\max} (G^2 + \sigma^2)
$$

$$
\leq -\frac{c_1 \eta_{\text{base}} E_{\min}}{2} \mathbb{E}[\|\nabla F(w^{(r)})\|^2] + \frac{c_1 \eta_{\text{base}} E_{\min} L^2}{S^{(r)}} \sum_{k \in \mathcal{A}^{(r)}} p_k \phi(\tau_k^{(r)}) \mathbb{E}[\|d_k^{(r)}\|^2]
$$

$$
+ c_2 \eta_{\text{base}} E_{\max} (G^2 + \sigma^2).
$$

We next bound the Local Variance and Global Variance terms using Lemmas B.3 and B.4. Lemma B.3 gives:

$$
\mathbb{E}[\|\Delta_k^{(r)}\|^2 \mid w^{(R_k^{(r)})}] \leq c_3 \eta_{\text{base}}^2 E_{\min}^2 \left( \mathbb{E}[\|\nabla F(w^{(R_k^{(r)})})\|^2] + G^2 + \sigma^2 \right)
$$

Using Lemma B.4, we have

$$
\text{(Global Variance)} \leq \frac{L}{2 S^{(r)}} \sum_{k \in \mathcal{A}^{(r)}} p_k \phi(\tau_k^{(r)}) \mathbb{E}[\|\Delta_k^{(r)}\|^2]
$$

$$
\leq \frac{L c_3 \eta_{\text{base}}^2 E_{\min}^2}{2 S^{(r)}} \sum_{k \in \mathcal{A}^{(r)}} p_k \phi(\tau_k^{(r)}) \left( \mathbb{E}[\|\nabla F(w^{(R_k^{(r)})})\|^2] + G^2 + \sigma^2 \right).
$$

Similarly, the Local Variance term is

$$\text{(Local Variance)} \leq \frac{c_3 \eta_{\text{base}}^2 E_{\min}^2}{2S^{(r)}} \sum_{k \in \mathcal{A}^{(r)}} p_k \phi(\tau_k^{(r)}) \left( \mathbb{E}[\|\nabla F(w^{(R_k^{(r)})})\|^2] + G^2 + \sigma^2 \right).$$

Combining these two terms and absorbing constants into $c_3 > 0$, we obtain

$$\text{(Global Variance)} + \text{(Local Variance)} \leq \frac{c_3 \eta_{\text{base}}^2 E_{\min}^2}{S^{(r)}} \sum_{k \in \mathcal{A}^{(r)}} p_k \phi(\tau_k^{(r)}) \left( \mathbb{E}[\|\nabla F(w^{(R_k^{(r)})})\|^2] + G^2 + \sigma^2 \right). \quad (10)$$

Using the smoothness relation above:

$$\|\nabla F(w^{(R_k^{(r)})})\|^2 \leq 2\|\nabla F(w^{(r)})\|^2 + 2L^2 \|d_k^{(r)}\|^2,$$

we bound the gradient term in (10) as

$$\frac{1}{S^{(r)}} \sum_{k \in \mathcal{A}^{(r)}} p_k \phi(\tau_k^{(r)}) \mathbb{E}[\|\nabla F(w^{(R_k^{(r)})})\|^2] \leq 2\mathbb{E}[\|\nabla F(w^{(r)})\|^2] + \frac{2L^2}{S^{(r)}} \sum_{k \in \mathcal{A}^{(r)}} p_k \phi(\tau_k^{(r)}) \mathbb{E}[\|d_k^{(r)}\|^2].$$

Substituting this to (10) and absorbing constants, we have

$$\text{(Global Variance)} + \text{(Local Variance)} \leq c_4 \eta_{\text{base}}^2 E_{\min}^2 \mathbb{E}[\|\nabla F(w^{(r)})\|^2] + \frac{c_5 \eta_{\text{base}}^2 E_{\min}^2 L^2}{S^{(r)}} \sum_{k \in \mathcal{A}^{(r)}} p_k \phi(\tau_k^{(r)}) \mathbb{E}[\|d_k^{(r)}\|^2]$$

$$+ c_6 \eta_{\text{base}}^2 E_{\min}^2 \left( G^2 + \sigma^2 \right),$$

for some constants $c_4, c_5, c_6 > 0$.

Combining (Progress), (Staleness), (Local Variance), and (Global Variance), we see that all drift-related contributions appear with a common structure

$$D_r := \frac{1}{S^{(r)}} \sum_{k \in \mathcal{A}^{(r)}} p_k \phi(\tau_k^{(r)}) \mathbb{E}[\|d_k^{(r)}\|^2],$$

multiplied by coefficients of order $\eta_{\text{base}} E_{\min} L^2$ and $\eta_{\text{base}}^2 E_{\min}^2 L^2$.

By Lemma B.2, the Staleness term is

$$D_r \leq \frac{1}{S^{(r)}} \sum_{k \in \mathcal{A}^{(r)}} p_k \phi(\tau_k^{(r)}) \tau_{\max} \sum_{t=0}^{r-1} \mathbb{E}[\|\Delta^{(t)}\|^2] = \tau_{\max} \sum_{t=0}^{r-1} \mathbb{E}[\|\Delta^{(t)}\|^2]$$

Lemmas B.3 and B.4 imply that each $\mathbb{E}[\|\Delta^{(t)}\|^2]$ is of order

$$\mathbb{E}[\|\Delta^{(t)}\|^2] \leq c_3 \eta_{\text{base}}^2 E_{\min}^2 \left( \frac{1}{S^{(t)}} \sum_{k \in \mathcal{A}^{(t)}} p_k \phi(\tau_k^{(t)}) \mathbb{E}[\|\nabla F(w^{(R_k^{(t)})})\|^2] + G^2 + \sigma^2 \right)$$

$$\leq c_3 \eta_{\text{base}}^2 E_{\min}^2 \left( \frac{1}{S^{(t)}} \sum_{k \in \mathcal{A}^{(t)}} p_k \phi(\tau_k^{(t)}) \left( 2\mathbb{E}[\|\nabla F(w^{(t)})\|^2] + 2L^2 \|d_k^{(t)}\|^2 \right) + G^2 + \sigma^2 \right)$$

$$\leq c_3 \eta_{\text{base}}^2 E_{\min}^2 \left( 2\mathbb{E}[\|\nabla F(w^{(t)})\|^2] + G^2 + \sigma^2 \right) + \frac{2c_3 \eta_{\text{base}}^2 E_{\min}^2 L^2}{S^{(t)}} \sum_{k \in \mathcal{A}^{(t)}} p_k \phi(\tau_k^{(t)}) \|d_k^{(t)}\|^2 \quad (11)$$

Define $a_s := \mathbb{E}[\|\Delta^{(s)}\|^2]$. Then we can write (11) as the recursive inequality

$$a_t \leq \alpha \mathbb{E}[\|\nabla F(w^{(t)})\|^2] + \beta(G^2 + \sigma^2) + \gamma \sum_{s=0}^{t-1} a_s$$

$$\leq \alpha \mathbb{E}[\|\nabla F(w^{(t)})\|^2] + \beta(G^2 + \sigma^2) + \gamma t \max_{0 \leq s \leq t} a_s,$$

where $\alpha = 2c_3\eta_{\text{base}}^2 E_{\text{min}}^2$, $\beta = c_3\eta_{\text{base}}^2 E_{\text{min}}^2$, and $\gamma = 2c_3\eta_{\text{base}}^2 E_{\text{min}}^2 L^2 \tau_{\text{max}}$. Let $M_t := \max_{0 \le s \le t} a_s$. Taking maximum on the left:

$$M_t \le \alpha \max_{s \le t} \mathbb{E}[\|\nabla F(w^{(s)})\|^2] + \beta(G^2 + \sigma^2) + \gamma R M_{t-1},$$

since $t \le R$. Rearranging the RHS term with $M_{t-1}$:

$$M_t \le \frac{1}{1 - \gamma R} \left( \alpha \max_{s \le t} \mathbb{E}[\|\nabla F(w^{(s)})\|^2] + \beta(G^2 + \sigma^2) \right)$$

For small enough $\eta_{\text{base}}$, the factor $\frac{1}{1-\gamma R}$ is a finite constant that depends only on $(L, \tau_{\text{max}}, R)$ and is independent of $k, t$. Absorb this into a new constant $C_7$, and focus back on $a_t \le M_t$:

$$a_t = \mathbb{E}[\|\Delta^{(t)}\|^2] \le c_7\eta_{\text{base}}^2 E_{\text{min}}^2 \left( \mathbb{E}[\|\nabla F(w^{(t)})\|^2] + L^2\tau_{\text{max}}^2 + G^2 + \sigma^2 \right),$$

where

$$\max_{s \le t} \|\nabla F(w^{(s)})\|^2 \le 2\|\nabla F(w^{(t)})\|^2 + 2L^2\tau_{\text{max}}^2.$$

We would like to note that all quantities here—$L$, $\tau_{\text{max}} = \lceil 1 + \gamma \rceil$, $G$, and $\sigma$—are dimensionless real scalars, so $L^2\tau_{\text{max}}^2$, $G^2$, and $\sigma^2$ are non-negative scalars of the same type and may be summed directly. The $L^2\tau_{\text{max}}^2$ term acts as a correction bounding gradient drift over the staleness window; a similar penalty structure appears in the asynchronous FL convergence bounds of FedBuff (Nguyen et al., 2022).

Then, the drift-related term can be bounded as

$$D_r \le \tau_{\text{max}} \sum_{t=0}^{r-1} \mathbb{E}[\|\Delta^{(t)}\|^2] \le c_7\tau_{\text{max}}\eta_{\text{base}}^2 E_{\text{min}}^2 \left( \sum_{t=0}^{r-1} \mathbb{E}[\|\nabla F(w^{(t)})\|^2] + rL^2\tau_{\text{max}}^2 + rG^2 + r\sigma^2 \right).$$

Plugging (Progress), (Staleness), (Local Variance), and (Global Variance), we have:

$$\begin{aligned}
\mathbb{E}[F(w^{(r+1)})] - \mathbb{E}[F(w^{(r)})] &\le \left( -\frac{c_1\eta_{\text{base}}E_{\text{min}}}{2} + c_4\eta_{\text{base}}^2 E_{\text{min}}^2 \right) \mathbb{E}[\|\nabla F(w^{(r)})\|^2] \\
&\quad + (c_2\eta_{\text{base}}E_{\text{min}} + c_6\eta_{\text{base}}^2 E_{\text{min}}^2)(G^2 + \sigma^2) \\
&\quad + \left( c_1\eta_{\text{base}}E_{\text{min}} + c_5\eta_{\text{base}}^2 E_{\text{min}}^2 + \frac{1}{2} \right) L^2 D_r \\
&\le \left( -\frac{c_1\eta_{\text{base}}E_{\text{min}}}{2} + c_4\eta_{\text{base}}^2 E_{\text{min}}^2 + A \right) \mathbb{E}[\|\nabla F(w^{(r)})\|^2] \\
&\quad + (c_2\eta_{\text{base}}E_{\text{min}} + c_6\eta_{\text{base}}^2 E_{\text{min}}^2 + Ar)(G^2 + \sigma^2) \\
&\quad + ArL^2\tau_{\text{max}}^2,
\end{aligned}$$

where

$$A = c_7\tau_{\text{max}}\eta_{\text{base}}^2 E_{\text{min}}^2 L^2 \left( c_1\eta_{\text{base}}E_{\text{min}} + c_5\eta_{\text{base}}^2 E_{\text{min}}^2 + \frac{1}{2} \right)$$

Summing over $r = 0, \ldots, R-1$, and telescoping the left-hand side gives:

$$\mathbb{E}[F(w^{(R)})] - \mathbb{E}[F(w^{(0)})] \leq \left(-\frac{c_1 \eta_{\text{base}} E_{\min}}{2} + c_4 \eta_{\text{base}}^2 E_{\min}^2\right) \sum_{r=0}^{R-1} \mathbb{E}[\|\nabla F(w^{(r)})\|^2] + A \sum_{r=0}^{R-1} \sum_{t=0}^{r-1} \mathbb{E}[\|\nabla F(w^{(t)})\|^2]$$

$$+ R\left(c_2 \eta_{\text{base}} E_{\min} + c_6 \eta_{\text{base}}^2 E_{\min}^2 + A \sum_{r=0}^{R-1} r\right)(G^2 + \sigma^2)$$

$$+ AL^2 \tau_{\max}^2 \sum_{r=0}^{R-1} r$$

$$\leq \left(-\frac{c_1 \eta_{\text{base}} E_{\min}}{2} + c_4 \eta_{\text{base}}^2 E_{\min}^2 + AR\right) \sum_{r=0}^{R-1} \mathbb{E}[\|\nabla F(w^{(r)})\|^2]$$

$$+ R\left(c_2 \eta_{\text{base}} E_{\min} + c_6 \eta_{\text{base}}^2 E_{\min}^2 + AR^2\right)(G^2 + \sigma^2)$$

$$+ AL^2 \tau_{\max}^2 R^2$$

$$\leq -c_8 \eta_{\text{base}} E_{\min} \sum_{r=0}^{R-1} \mathbb{E}[\|\nabla F(w^{(r)})\|^2] + c_9 R \eta_{\text{base}}^2 E_{\min}^2 (L^2 \tau_{\max}^2 + G^2 + \sigma^2)$$

where there exists a step-size constant small enough to guarantee descent condition.

Using $F(w^{(R)}) \geq F^*$, we rearrange

$$\frac{1}{R}\sum_{r=0}^{R-1} \mathbb{E}[\|\nabla F(w^{(r)})\|^2] \leq C_0 \frac{\mathbb{E}[F(w^{(0)})] - F^*}{\eta_{\text{base}} E_{\min} R} + C_1 \eta_{\text{base}} E_{\min}(L^2 \tau_{\max}^2 + G^2 + \sigma^2),$$

where $C_0 = \frac{1}{c_8}$ and $C_1 = \frac{c_9}{c_8}$. $\qquad\qquad\square$

## C. Sensitivity of Staleness Decay Functions

We sweep the decay parameter $\beta \in \{0.25, 0.5, 1.0, 2.0\}$ under both harmonic $\phi(\tau) = (1 + \beta\tau)^{-1}$ and exponential $\phi(\tau) = \exp(-\beta\tau)$ decay, holding all other settings fixed. Table 7 reports the average normalized weight assigned to late updates, alongside maximum accuracy and time to 90% accuracy. Smaller $\beta$ values preserve more stale-update information, while larger $\beta$ values penalize delayed updates more strongly. Among the tested settings, the harmonic staleness function with $\beta = 0.5$ provides the best trade-off: it shows a competitive peak accuracy (only lagging by 0.23% to the $\beta = 0.25$ setting) among the stable configurations and achieves the fastest time to 90% accuracy.

*Table 7.* Sensitivity to the staleness weighting function and decay parameter $\beta$ on non-IID MNIST (Dirichlet $\alpha_{\text{dir}} = 0.5$). Average normalized staleness weight measures the effective contribution assigned to late updates.

| $\beta$ | *Harmonic decay* | | | *Exponential decay* | | |
|---|---|---|---|---|---|---|
| | Avg. Weight | Max-A↑ | Time-to-$A^\star$ ↓ | Avg. Weight | Max-A↑ | Time-to-$A^\star$ ↓ |
| 0.25 | 0.177 | **96.27** | 73.26 | 0.173 | **95.11** | 72.19 |
| 0.50 | 0.152 | 96.04 | **67.98** | 0.140 | 94.91 | **70.60** |
| 1.00 | 0.118 | 94.91 | 75.04 | 0.090 | 94.74 | 76.17 |
| 2.00 | 0.082 | 92.45 | 81.18 | 0.035 | 92.08 | 89.65 |

## D. Additional Details for Large-Scale Experiments

This appendix complements Section 6.1 by providing additional details on the experimental setup and per facility-level diagnostics for the cross-facility LLaMA2-7B experiment.

### D.1. Experimental Setup

We configure two FEDQUEUE runs (FEDQUEUE-1 and FEDQUEUE-2) with synchronization horizon and safety buffer set to $(T_{\text{sync}}, \delta) = (20, 1)$ minutes and $(T_{\text{sync}}, \delta) = (40, 2)$ minutes respectively. The rest of the parameters are identical

across the two FEDQUEUE runs. All methods use identical per-round resource requests and training hyperparameters. The configurations for the real-world cross-facility deployment and FEDQUEUE-specific parameters are presented in Table 8.

*Table 8.* **Training protocols.**

| Configuration | Value |
|---|---|
| Model / tuning | LLaMA2-7B / full fine-tuning |
| Sequence length | 512 |
| Dataset | Partition by task categories (non-IID) |
| Optimizer / LR | AdamW / fixed and fine-tuned LR |
| Evaluation | After each aggregation |
| Wall-clock budget | 17,000 seconds |
| FEDQUEUE-*specific parameters:* | |
| $T_{\text{sync}}$ | 20 min for FEDQUEUE-1; 40 min for FEDQUEUE-2 |
| Safety buffer $\delta$ | 1 min for FEDQUEUE-1; 2 min for FEDQUEUE-2 |
| EWMA rate $\alpha$ | 0.5 |
| Staleness weight $\phi(\tau)$ | harmonic with $\beta = 0.5$ |
| Initial budget $E_k^{(1)}$ | 20 training steps |

## D.2. Per Facility-level Diagnostics

While the main text reports consolidated time-to-quality metrics (Figure 2 and Table 1), Figure 5 presents detailed loss trajectories for each method, showing both individual facility and global model performances.

Several patterns emerge from the facility-level view. FedAvg maintains relatively synchronized updates across facilities, but this synchronization comes at the cost of slower overall progress due to stragglers. FedAsync shows more rapid early progress, but ultimately plateaus at a higher final loss, suggesting that uncontrolled asynchrony may not be ideally in this real-world deployment case. In addition, FedCompass achieves competitive early performance as well through static queue profiling, but its reliance on fixed estimates causes performance degradation as queue conditions shift over the 5-hour experimental window. Finally, both FEDQUEUE configurations demonstrate more balanced facility utilization while achieving superior final global models, with FEDQUEUE-2's longer synchronization horizon (40 minutes) enabling better adaptation to queue variability and ultimately reaching the lowest final loss.

Beyond the loss trajectories, we also report the realized range of local-step budgets across rounds and facilities. The derived budgets $E_k^{(r)} = \left\lfloor c_k J_k^{(r)} \right\rfloor$ ranged from $E_{\min} = 20$ to $E_{\max} = 72$ steps under FEDQUEUE-1 and from $E_{\min} = 20$ to $E_{\max} = 176$ steps under FEDQUEUE-2. The $E_{\min} = 20$ step floor is conservative with respect to Theorem 5.5 where larger $E_{\min}$ tightens the optimization error term while leaving the bias term unchanged.

# E. Controlled Experiments and Extended Results

This appendix provides (i) the complete controlled-experiment configuration and (ii) extended results that validate Lemma 5.4 under synthetic queue dynamics. The main text in Section 6.2 reports a compact subset (time-to-quality, expected delay, max delay, Data movement ratio, total local steps); here we include the full sweeps over queue variability, admission tolerance, predictor quality, and safety-buffer stress tests, plus ablations.

## E.1. Experimental Overview

We use synthetic queue processes which let us vary queue variance, prediction quality, and admission thresholds in a reproducible way, directly probing the assumptions behind Lemma 5.4.

Table 9 records the dataset/model/optimization and evaluation configuration used for all controlled experiments. Unless stated otherwise, all methods share the same learning workload, compute settings, and evaluation protocol; differences are restricted to orchestration, budgeting, admission, and aggregation.

**Baselines.** We compare FEDQUEUE against:

- **FedAvg** (McMahan et al., 2017): synchronous FL with fixed local work per round and blocking aggregation.
- **FedAsync** (Xie et al., 2020): fully asynchronous aggregation with a staleness weight $w(\tau) = (1 + \tau)^{-1}$.
- **FedBuff** (Nguyen et al., 2022): buffered asynchronous FL with a fixed buffer size.
- **FedCompass** (Li et al., 2024): a system-aware baseline that allocates local work using static throughput profiling

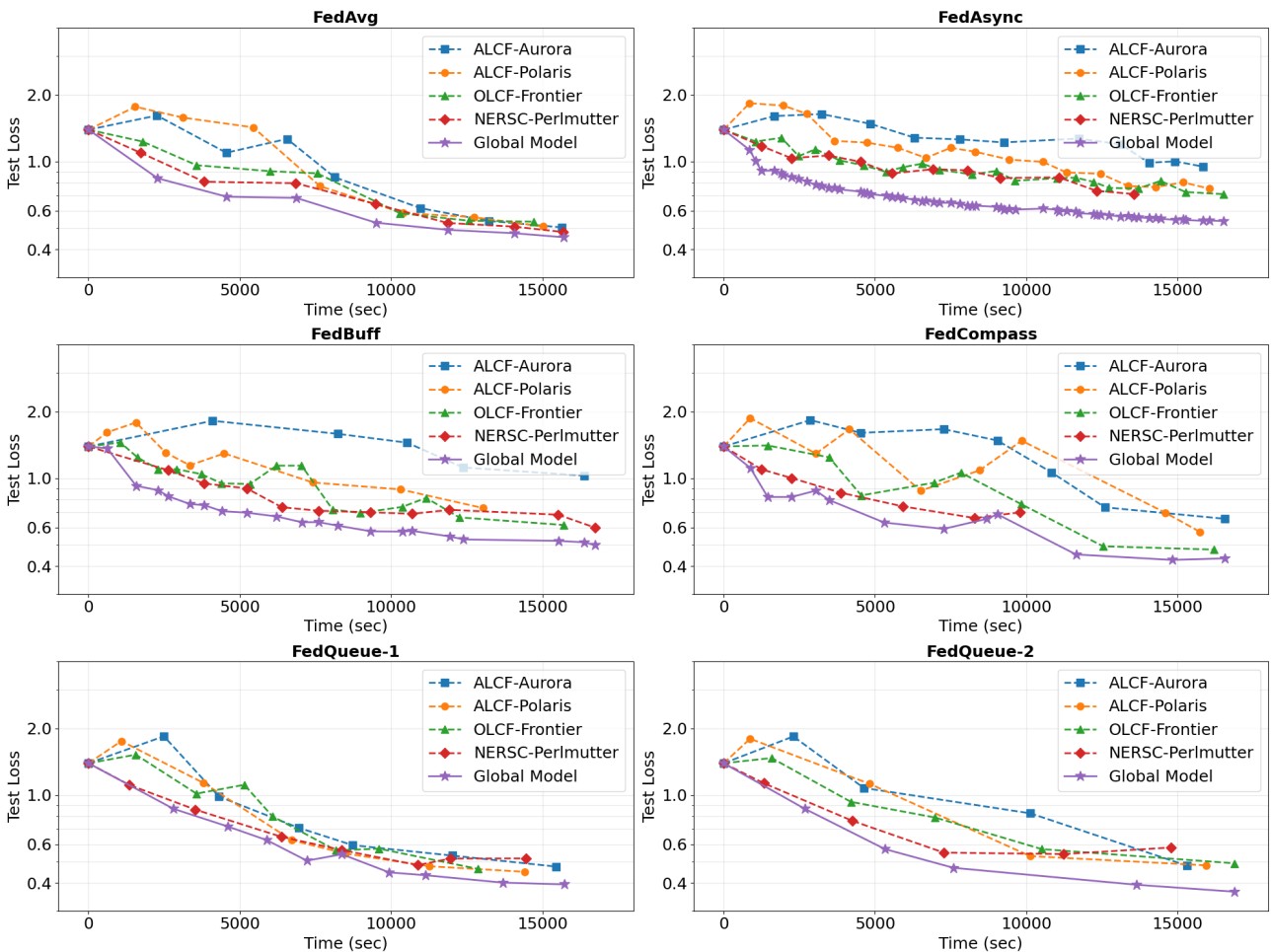

*Figure 5.* **Facility-level loss trajectories for all methods.** Each panel shows test loss over time for individual facilities (colored dashed lines) and the federated global model (purple solid line). FEDQUEUE configurations achieve better balance between facility utilization and global model quality.

without adapting to queue dynamics.

**Metrics.** We report:

- **Time-to-quality:** training loss / validation metric versus elapsed time to reach a target metric.
- **Empirical probability of client arrival beyond the cutoff:** $\mathbb{P}(t_k^r > T_{\text{sync}}, \forall(k, r))$
- **Normalized expectation of delay:** $\hat{\mathbb{E}}_d = \mathbb{E}\left[\frac{t_k}{T_{\text{sync}}} \,\middle|\, t_k > T_{\text{sync}}\right]$,
- **Normalized maximum delay:** $R_d = \max_{k,r}\left(\frac{t_k}{T_{\text{sync}}}\right)$

**Dataset and Partitioning.** We use the MNIST handwritten digit classification benchmark with a custom data loader. Unless otherwise stated, each client $k$ receives a disjoint training partition and all clients share a common held-out test set for evaluation. We consider (i) IID partitions and (ii) non-IID partitions generated by a Dirichlet allocation with concentration parameter $\alpha_{\text{dir}} = 0.5$. Each client evaluates on the shared test set to ensure comparable time-to-quality curves across methods.

**Model.** We train a lightweight CNN composed of two $3 \times 3$ convolution layers with channel sizes $(32, 64)$ and padding 1, each followed by $2 \times 2$ max pooling, then a fully connected layer $64 \cdot 7 \cdot 7 \rightarrow 128$, dropout $p = 0.5$, and a final 10-way linear classifier. ReLU activations are used throughout. The objective is cross-entropy and we report test accuracy as the primary quality metric.

**Hyperparameter Optimization.** We performed a targeted hyperparameter sweep focusing on the learning rate and batch size, while holding all other training settings fixed to isolate their effects. The learning rate was explored over the set $\{0.001, 0.003, 0.005, 0.0001, 0.0005\}$, and the batch size over $\{32, 64, 96, 128\}$. Each configuration was evaluated using the same training protocol and validation metric, and the optimal setting was selected based on validation accuracy. This search identified a learning rate of $0.003$ with a batch size of $64$ as the best-performing combination for our experiments. To support reproducibility, Table 9 summarizes the core hyperparameters.

**Baseline Implementation.** We implemented all the reported baseline methods using APPFL (Ryu et al., 2022), and varied the respective hyperparameters according to Table 9 to simulate the impact of queue variance, and admission control.

*Table 9.* **Controlled experiment configuration and hyperparameters.** This table consolidates the controlled-workload specification (MNIST + CNN) and the concrete run configuration/hyperparameters used in Appendix C sweeps (queue variability, admission tolerance, and prediction quality).

| Parameter | Value | Role / notes |
|---|---|---|
| **Workload (data/model/metric)** | | |
| Dataset | MNIST | Custom data loader; shared held-out test set across clients. |
| Partition | iid / non-iid | Non-iid via Dirichlet allocation. |
| data.alpha | 0.5 | Dirichlet concentration (for non-iid). |
| Model | SimpleCNN | 2 conv ($3{\times}3$) with channels (32,64), max-pool, FC $64 \cdot 7 \cdot 7 \rightarrow 128$, dropout 0.5, 10-way head. |
| loss.name | CELoss | Cross-entropy objective. |
| **Federated protocol (shared across baselines)** | | |
| seed | 42 | Reproducibility seed. |
| num_clients | 4 | Number of participating clients. |
| num_rounds | 50 | Total FL communication rounds. |
| gpus | cuda:0,1,2,3 | GPU devices assigned to clients (local testbed / simulation). |
| batch_size | 64 | Client runtime config. |
| Optimizer | Adam | Client runtime config. |
| local_steps | 100 | Client runtime config (per admitted update unless otherwise specified). |
| **FEDQUEUE orchestration / queue model knobs (swept in Appendix C)** | | |
| algo.name | fedqueue | Algorithm selector (fedqueue/fedavg/fedasync/fedbuff/fedcompass). |
| algo.broadcast_when | immediate / next_round | Broadcast timing policy. |
| algo.delay_mode | simulate / sleep | Delay handling (simulation vs. wall-clock sleeping). |
| Tsync | 10.0 | Target sync window ($T_{\mathrm{sync}}$). |
| q_init | 2.0 | Initial delay prior for admission control. |
| gamma | 0.2 | Slack fraction on $T_{\mathrm{sync}}$ for admission windowing. |
| delta | 2.0 | Safety buffer inside budget $J_k$ (maps to safety-buffer scaling). |
| alpha | 0.5 | EWMA smoothing for delay estimate $\hat{q}_k$. |
| warmup_steps | 10 | Warm-up steps to estimate throughput $c_k$. |
| sim_queue | lognormal | Queue model (fixed / lognormal). |
| queue_fixed | 0.5,1.5,2.4,6 | Fixed delays per client (if sim_queue=fixed). |
| queue_means | 1.5,2.5,3.5,4.5 | Lognormal means per client. |
| queue_rho | 0.4 | Lognormal standard deviation. |
| slowdown | 1.0,1.0,1.0,1.0 | Compute slowdown multipliers per client. |
| staleness_mode | harmonic | Staleness decay (harmonic / exp). |
| staleness_beta | 0.5 | Staleness decay strength. |
| admission_horizon | horizon | Admission policy (horizon / all). |
| client_weight_mode | equal | Client weight mode (equal / data size). |
| lr_base | 0.003 | Base learning rate. |
| fedavg.num_local_steps | 67,155,147,15 | Local steps per client under FedAvg path. |
| **Global AsyncFL settings** | | |
| num_local_steps | 155 | Number of local update steps per client |
| staleness_fn | polynomial | Staleness decay function (constant / polynomial / hinge) |
| staleness_fn_kwargs | {a=1.0} | Polynomial decay coefficient |
| alpha | 0.5 | Staleness scaling factor |
| optimize_memory | true | Enables memory-optimized aggregation |
| **FedBuff overrides** | | |
| $K$ | 2/3/4 | Buffer size (number of updates before aggregation) |
| **FedCompass overrides** | | |
| staleness_fn | polynomial | Staleness-aware weighting function |
| staleness_fn_kwargs | {} | Uses default polynomial parameters |
| alpha | 0.5 | Staleness scaling factor |
| max_local_steps | 200 | Upper bound on adaptive local steps |
| min_local_steps | 20 | Lower bound on adaptive local steps |
| speed_momentum | 0.6 | Momentum term for client speed estimation |
| latest_time_factor | 1.1 | latest allowable arrival window |

### E.2. Queue Knobs and Sweep Design

We vary three parameters aligned with Lemma 5.4: (i) queue variability via the sub-Gaussian scale $\rho_k$, (ii) admission tolerance $\gamma$ (implying $\tau_{\max} = \lceil 1 + \gamma \rceil$), and (iii) prediction quality via EWMA rate $\alpha$ controlling the error $e_k^{(r)} = q_k^{(r)} - \hat{q}_k^{(r)}$ and thus the safety-buffer requirement. All sweeps are reported using pre-specified artifacts (Figures 6, 7, 8 and Table 10).

**Rho sweep (staleness concentration).** Figure 6 illustrates the empirical CDF of client arrivals beyond $T_{\mathrm{sync}}$ under increasing queue variability for both IID and non-IID data partitions. Under low queue variance ($\rho_k = 0.1$), the CDF rises sharply: over 90% arrive before 8 seconds. As queue variability increases, the delay distribution becomes progressively heavier tailed. For $\rho_k = 0.5$, the 90th percentile shifts to roughly 10–12 seconds, while for $\rho_k = 0.9$ it extends beyond 15 seconds. The vertical dashed line marking $T_{\mathrm{sync}}$ highlights that although the majority of updates still arrive before the cutoff, the tail mass near the boundary increases from only a few percent at $\rho_k = 0.1$ to well over 10% at $\rho_k = 0.9$. This trend is consistent across both IID and non-IID regimes, with the non-IID case exhibiting a slightly heavier tail.

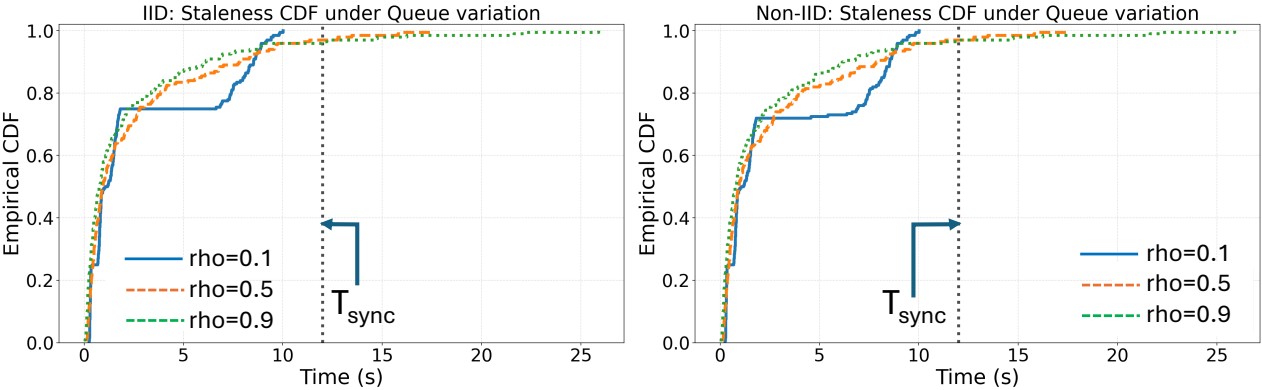

*Figure 6.* **Staleness under queue variance.** Empirical CDF of clients arrived beyond $T_{\mathrm{sync}}$ under a sweep of $\rho$. This plot audits the bounded-staleness behavior induced by $\rho$.

**Gamma sweep (time-to-quality).** Figure 7 shows how tightening/relaxing admission tolerance $\gamma$ affects the time-to-quality and convergence across baselines. Across all $\gamma$ values, FEDQUEUE consistently reaches high accuracy earlier than competing baselines. In the IID case, FEDQUEUE with $\gamma = 0.9$ surpasses 98% validation accuracy within roughly 30–40 seconds, whereas FedBuff and FedCompass require approximately 60–90 seconds to reach the same level. Under non-IID data, the gap widens: FEDQUEUE reaches 95% accuracy in about 70–90 seconds, while FedBuff and FedCompass typically require 150–200 seconds. Increasing $\gamma$ accelerates early convergence for all methods; however, FEDQUEUE exhibits the strongest sensitivity.

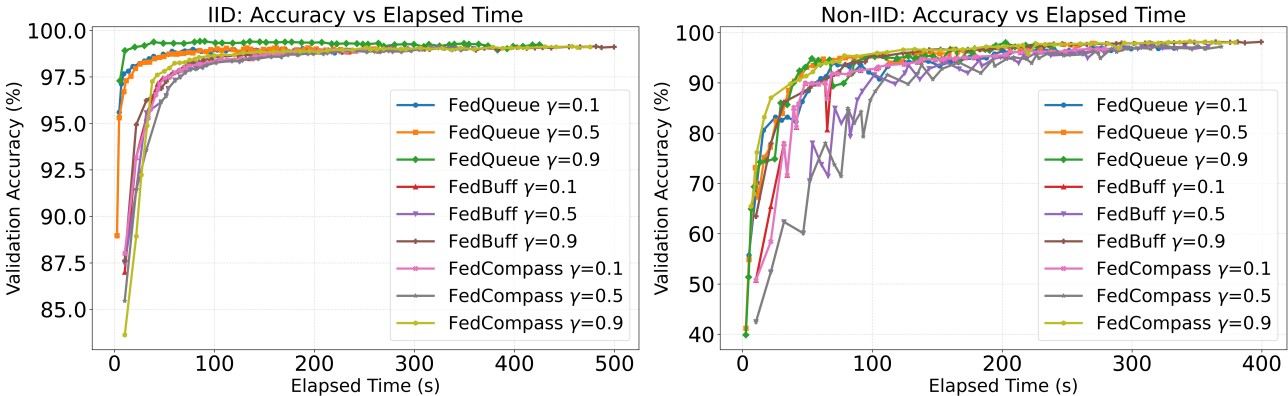

*Figure 7.* **Time-to-quality varying admission parameter.** Validation accuracy versus elapsed time under a sweep of admission parameter $\gamma$ (low/medium/high or a multi-level sweep). FEDQUEUE consistently converges faster and reaches the highest accuracy compared to the baselines for both IID and non-IID cases.

**Alpha sweep (staleness concentration).** Figure 8 shows how tightening/relaxing EWMA rate $\alpha$ affects the empirical CDF of clients arriving after $T_{\mathrm{sync}}$. Both $\alpha = 0.1$ (slow tracking) and $\alpha = 1.0$ (aggressive tracking) exhibit samples

extending beyond the synchronization horizon $T_{\text{sync}}$, indicating that overly conservative smoothing and overly reactive delay estimation can each induce late client arrivals. In contrast, the moderate choice $\alpha = 0.5$ ensures all client updates arriving within the synchronization window $T_{\text{sync}}$, which is consistent across both IID and non-IID settings.

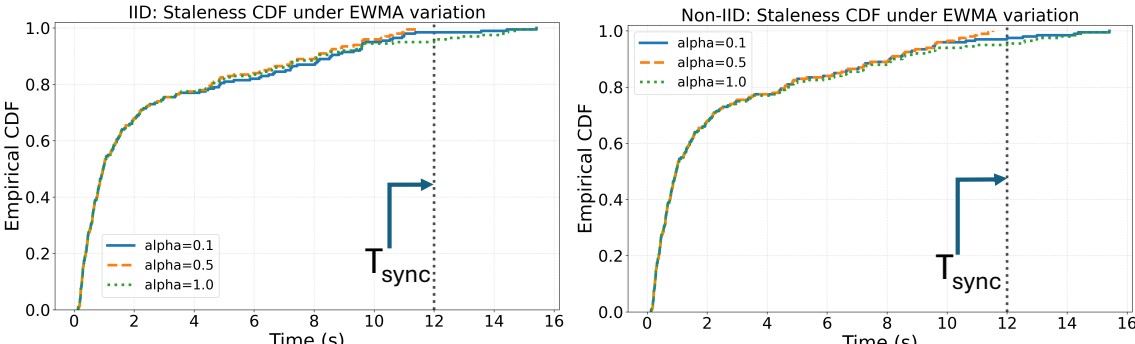

*Figure 8.* **Staleness under EWMA rate ($\alpha$) variation.** Empirical CDF of clients arrived beyond $T_{\text{sync}}$ under a sweep of $\alpha$. This plot audits the bounded-staleness behavior induced by $\alpha$.

**Bound verification grid.** Table 10 reports violation frequencies, and maximum observed delay across a representative grid of $(\rho_k, \gamma, \alpha)$ settings, together with time-to-target.

*Table 10.* **Bound verification grid (controlled).** For each setting in a representative $(\rho_k, \gamma, \alpha)$ grid, we report the empirical probability of client arrival beyond the cutoff, $\mathbb{P}$, Normalized expectation of delay, $\hat{\mathbb{E}}_d$, Normalized maximum delay, $R_d$, and the wall-clock time to reach target accuracy $A^\star$ (95%).

| $\rho$ | $\gamma$ | $\alpha$ | $\mathbb{P}\downarrow$ | $\hat{\mathbb{E}}_d\downarrow$ | $R_d\downarrow$ | Time-to-$A^\star$ (s)$\downarrow$ |
|---|---|---|---|---|---|---|
| 0.1 | 4 | 0.5 | 0.000 | - | - | 156.65 |
| 0.5 | 4 | 0.5 | 0.020 | 1.14 | 1.18 | 159.31 |
| 0.9 | 4 | 0.5 | 0.035 | 1.38 | 1.52 | 163.95 |
| 0.1 | 1 | 0.5 | 0.035 | 1.32 | 1.56 | 146.66 |
| 0.1 | 2 | 0.5 | 0.015 | 1.06 | 1.17 | 151.89 |
| 0.1 | 4 | 0.5 | 0.000 | - | - | 156.65 |
| 0.1 | 4 | 0.1 | 0.015 | 1.32 | 1.44 | 162.73 |
| 0.1 | 4 | 0.5 | 0.000 | - | - | 156.65 |
| 0.1 | 4 | 1.0 | 0.020 | 1.34 | 1.46 | 164.62 |

### E.3. Scalability Results

We perform the controlled experiments of with $K = 8$ and $K = 12$, keeping high variance of the queue, $\rho = 0.9$, and the non-IID partition. All other settings match Table 4 in the main text. Table 11 reports maximum accuracy, time to reach 85% accuracy, and total local steps ($\#E_k$).

Across both scales, FEDQUEUE reaches the 85% target fastest while using the fewest local steps. Comparing with the $K = 4$ result in Table 4, the threshold was lowered to 85% here because non-IID training at larger $K$ converges more slowly, the relative gap over FedAvg holds steady (1.2× with $K = 8$ and 1.1× with $K = 12$) and the gap over FedCompass widens (1.7× with $K = 8$ and 1.5× with $K = 12$). Local step counts also remain the lowest among all baselines as $K$ grows, indicating that the queue-aware budgeting continues to allocate work efficiently as the client pool expands.

### E.4. Stronger Non-IID Partition Results

We evluate FEDQUEUE under more severe data heterogeneity. In particular, we partition MNIST using a Dirichlet allocation with concentration parameter $\alpha_{\text{dir}} = 0.1$. All other settings are fixed. We compare FEDQUEUE against FedAvg and FedCompass, the two strongest synchronous and asynchronous baselines in Table 12.

FEDQUEUE reaches the 85% target 1.8× faster than FedAvg and 3.2× faster than FedCompass, using roughly half the

*Table 11.* Scalability under high queue variance ($\rho = 0.9$) and non-IID partitioning. FEDQUEUE achieves the fastest time-to-85% and the fewest local steps with both $K = 8$ and $K = 12$.

| Method | $K = 8$ | | | $K = 12$ | | |
|---|---|---|---|---|---|---|
| | Max-A↑ | Time-to-$A^*$ ↓ | $\#E_k$ ↓ | Max-A↑ | Time-to-$A^*$ ↓ | $\#E_k$ ↓ |
| FedAvg | **97.4** | 106.6 | 4798 | **97.3** | 94.4 | 6682 |
| FedAsync | 90.5 | 417.3 | 19680 | 85.6 | 386.2 | 18140 |
| FedBuff | 94.3 | 141.9 | 7440 | 91.4 | 142.0 | 10080 |
| FedCompass | 95.8 | 153.2 | 9281 | 91.7 | 126.5 | 8991 |
| FEDQUEUE | 97.0 | **91.2** | **4288** | 96.5 | **86.7** | **5628** |

local steps of FedAvg and less than a third of FedCompass. FedCompass attains the highest final accuracy at the cost of substantially more wall-clock time and local computation. The widening time-to-target gap relative to the milder $\alpha_{\mathrm{dir}} = 0.5$ is consistent with the pattern that FEDQUEUE's advantage grows as data heterogeneity intensifies.

*Table 12.* Stronger non-IID partitioning with Dirichlet $\alpha_{\mathrm{dir}} = 0.1$. Max accuracy, time-to-85%, and local steps ($\#E_k$).

| Method | Max-A↑ | Time-to-$A^*$ ↓ | $\#E_k$ ↓ |
|---|---|---|---|
| FedAvg | 86.0 | 140.6 | 5760 |
| FedCompass | **91.1** | 252.7 | 10386 |
| FEDQUEUE | 88.7 | **78.6** | **3136** |

