# OpenReview forum: "FedQueue: Queue-Aware Federated Learning for Cross-Facility HPC Training"
_ICML.cc/2026/Conference — ICML 2026 regular_

### Official Review · Reviewer_Fejh · 2026-03-07

**Soundness:** 2
**Presentation:** 3
**Significance:** 3
**Originality:** 3
**Overall Recommendation:** 4
**Confidence:** 3

**Summary:**

The paper introduces FedQueue, a semi-asynchronous federated learning protocol explicitly designed to mitigate the impact of unpredictable queue delays in shared High-Performance Computing (HPC) environments. To prevent slow or queued clients from causing extreme model staleness, the authors propose a system that integrates an Exponentially Weighted Moving Average (EWMA) queue predictor with an adaptive admission control mechanism. This system dynamically determines which clients' updates should be accepted in a given round based on a calculated safety buffer. The authors provide theoretical convergence guarantees and validate their approach through both controlled synthetic experiments and a highly complex, real-world deployment using APPFL and GLOBUS to fine-tune a LLaMA2-7B model across geographically distributed, heterogeneous HPC facilities.

**Compliance With Llm Reviewing Policy:**

Affirmed.

**Final Justification:**

I believe the work presented in the paper matches the score I assigned in the review.

**Key Questions For Authors:**

**Q1: Dimensional Inconsistency in Appendix B.5**
In the theoretical derivation, a staleness-related term is directly added to the squared gradient norm. These terms appear to be dimensionally incompatible without an appropriate scaling factor (e.g., step-size related). Could the authors clarify how this dimensional mismatch is reconciled mathematically and physically?

**Q2: Assumptions on HPC Queue Distributions**
The theoretical proofs and synthetic data seem to rely on the assumption that queue delays or prediction errors follow a sub-Gaussian distribution. Since real-world HPC queues often exhibit more complex, heavy-tailed behaviors, could the authors provide empirical evidence or citations supporting this assumption?

**Q3: Experimental Scalability**
The current evaluations are based on a 4-client setup. This scale may not fully capture the resource contention, staleness, and queue backlogs typical of larger cross-silo federations. Can the authors provide synthetic or emulated results for a larger number of clients?

**Q4: Choice of Queue Predictor**
The system employs an EWMA predictor. Given the highly dynamic and complex nature of HPC schedulers, why was EWMA chosen over more sophisticated prediction methods (e.g., ML-based approaches)?

**Limitations:**

Yes, the authors have included a brief discussion of limitations in the concluding remarks, acknowledging the sub-Gaussian queue assumption and the lack of explicit modeling for job failures. To further strengthen the paper, I recommend expanding this section to provide a more comprehensive view of the system's practical boundaries.

Constructive suggestions for improvement:
1. **Practical Implications of Job Failures:** While the authors rightly note that job failures result in lost updates, it would be highly valuable to discuss how this impacts the system operationally.
2. **Scale of Empirical Validation:** It would be helpful to explicitly acknowledge the scale of the current controlled experiments (e.g., the 4-client setup). Discussing how resource contention, extreme delays, and staleness bounds might evolve in larger, more congested cross-silo federations would provide excellent context for future research directions.
3. **Queue Prediction Enhancements:** While the EWMA predictor provides a lightweight and interpretable baseline, it is inherently a simple linear filter. It would be beneficial to acknowledge its limitations in capturing the highly non-linear, bursty dynamics typical of production HPC schedulers. Discussing the potential to integrate more advanced, machine learning-based queue prediction methods in future work would add valuable depth.

**Strengths And Weaknesses:**

**1. Soundness**
* **Strengths:** The empirical deployment of FedQueue to fine-tune a LLaMA2-7B model across real, distributed HPC centers is methodologically impressive. It serves as a strong "proof of existence" that the system architecture functions under real-world infrastructure constraints. Furthermore, the ablation studies are well-designed, effectively isolating how individual components (like the EWMA predictor and the staleness decay function) contribute to overall convergence.
* **Weaknesses:** The paper suffers from critical flaws in both its theoretical rigor and systemic assumptions. Dimensional Inconsistency in Proofs: The theoretical derivation contains a fundamental dimensional mismatch that undermines the convergence guarantees. The authors directly add an unscaled staleness term to the squared gradient norm. Physically, this staleness term must be scaled by the learning rate squared (or a similar step-size factor) to be dimensionally compatible. Fragile Distribution Assumptions: The proofs rely heavily on the assumption that queue-prediction errors follow a sub-Gaussian distribution. However, the authors provide no supporting literature or empirical evidence to justify that real-world HPC queues conform to this assumption. Lack of Fault Tolerance: The protocol design overlooks common HPC node failures (e.g., out-of-memory errors, preemption, hardware crashes).

**2. Presentation**
* **Strengths:** The submission is clearly written, and the overall narrative is easy to follow. The authors do an excellent job of motivating the problem, clearly outlining why traditional synchronous and asynchronous FL baselines fail when confronted with HPC job schedulers. The system architecture diagrams and logical flow make the framework highly accessible.
* **Weaknesses:** The presentation glosses over the fact that their chosen queue predictor (EWMA) is a simple, backward-looking filter that is fundamentally incapable of capturing the step-change dynamics or maintenance-window anomalies typical of production schedulers like Slurm or PBS.

**3. Significance**
* **Strengths:** The paper addresses a highly important and relevant problem. As federated learning moves toward foundation models that require multi-node GPU clusters, mitigating unpredictable scheduler delays is a critical roadblock. Providing a framework that allows FL to gracefully co-exist with standard HPC queuing systems offers immense practical utility and is very likely to influence how practitioners deploy cross-silo FL in the future.
* **Weaknesses:** The significance of the empirical claims regarding "bounded staleness" is heavily undermined by the insufficient scale of the controlled experiments. Evaluating the system on a strictly 4-client setup is a "toy scale" configuration that completely masks the complexities of heavy queue backlogs.  Whether the system can actually maintain efficiency and bound staleness in a true cross-silo federation involving dozens of highly congested institutional clients.

**4. Originality**
* **Strengths:** The work demonstrates high originality through a creative combination of existing ideas. By bridging systems engineering (queue prediction and admission control) with machine learning optimization (asynchronous FL aggregation), the authors highlight a critical blind spot in current FL literature. Applying these combined concepts to a heavy-weight, real-world use case (LLM fine-tuning across distinct supercomputing centers) pushes the boundaries of applied federated learning.
* **Weaknesses:** While the cross-disciplinary combination is novel, the individual algorithmic components are somewhat incremental. The EWMA predictor is a standard, rudimentary forecasting tool, and the staleness decay mechanisms heavily borrow from existing asynchronous FL literature. The originality lies primarily in the system's integration and application rather than introducing a groundbreaking new optimization theory.

---

> ### Author Rebuttal · Authors · 2026-03-31
>
> Dear Reviewer Fejh,
>
> We thank you for the detailed review and positive assessment of our real-world deployment, presentation clarity, and originality. Please find our responses below.
>
> **W1. & Q1. & Q2. theoretical results and assumption.**
>
> *Dimensional inconsistency (W1, Q1).* In our framework all quantities, $L, \tau_{\max}, G, \sigma$, are dimensionless real scalars. $L$ is the Lipschitz constant, $\tau_{\max} = \left\lceil 1 + \gamma \right\rceil$ is a dimensionless integer, and $G^2$, $\sigma^2$ are dimensionless scalar bounds on squared norms. Consequently $L^2\tau^2_\max$, $G^2$, and $\sigma^2$ are all non-negative real numbers of the same type, and their sum in the bias term is mathematically well-formed. We believe that the specific line in question in Appendix B.5 bounds $\max_{s \leq t} ||\nabla F(w^{(s)})||^2$ by $2||\nabla F(w^{(t)})||^2 +2L^2\tau^2_\max$ where $2L^2\tau^2_\max$ serves as a correction term bounding gradient norm drift. This is dimensionally consistent because all quantities are dimensionless scalars. We will add a clarifying remark to Appendix B.5. We point out that a similar $L^2\tau^2_\max$ penalty structure appears in asynchronous FL convergence bounds in Nguyen et al. (2022).
>
> *Sub-Gaussian assumption (W1, Q2).* This concern is shared with Reviewer eC2p. Please refer to our detailed response there, which includes empirical characterization of prediction errors across all four facilities. In summary: the assumption governs prediction errors $e_k^{(r)}$, not raw queue delays. After removing outlier rounds corresponding to step-change queue events, cleaned residuals show near-zero means and standard deviations of 39-114 sec. (3-10% of $T_{sync}$). Most importantly, the practical consequence, bounded staleness, is directly validated by Table 3: maximum observed staleness is consistent with $\tau_\max$ from Lemma 5.4.
>
> *Fault tolerance (W1).* We acknowledge fault tolerance is not currently modeled. In the current design, the server cannot distinguish a failed job (e.g., out-of-memory or hardware crashes) from a late arrival — a permanently failed job results in a lost update, noted as a limitation in Section 7. Implementing failure detection and automatic resubmission is a valuable future extension. For preempted jobs, most HPC schedulers (Slurm, PBS) support automatic requeue; a requeued job that completes returns its update with the appropriate staleness weight via the buffering mechanism. We will expand the fault tolerance discussion in the revision.
>
> **W2. & Q4. EWMA limitations and predictor choice.**
> We acknowledge EWMA is a backward-looking filter limited in capturing abrupt step-change dynamics or maintenance-window anomalies. Importantly, our real-world experiments directly demonstrate this limitation and FedQueue's robustness to it: the 1-2 outlier rounds per facility correspond precisely to step-change events where queue delays spiked beyond EWMA's anticipation. Despite these failures, the buffering mechanism handled each gracefully — late updates were deferred and incorporated with staleness weights, and maximum observed staleness remained bounded (Table 3). For scheduled maintenance windows, HPC facilities typically announce these in advance, allowing practitioners to avoid scheduling FL during disruptions. Our convergence analysis is prediction-agnostic, allowing substitution of ML-based predictors directly. One promising future direction is querying scheduler state (e.g. queue depth and pending jobs) before job submission to detect impending step-changes proactively. We will add this as future work and expand EWMA limitations in the revision.
>
> **W3. & Q3. scalability.**
> This comment is shared across all three reviewers. Please refer to our detailed response to Reviewer JJ4s's W7 which includes new controlled experiments with K=8 and K=12 clients. In summary, FedQueue maintains consistent speedup across K=8 and K=12 under non-IID partitioning: 1.2x-1.1x over FedAvg, 1.7x-1.5x over FedCompass, and 4.6x-4.5x over FedAsync, consistent with our theoretical prediction. We also observe stable convergence and bounded staleness at both K=8 and K=12, consistent with Lemma 5.4.
>
> **W4. originality.**
> We appreciate the balanced assessment and agree with the characterization. The originality of FedQueue lies in identifying and formalizing scheduler-induced admission delay as a first-class signal in cross-facility HPC FL, and the principled integration of prediction, adaptive budgeting, admission control, and staleness-aware aggregation into a cohesive system with provable guarantees. We will clarify this positioning in the revision.
>
> **L1. expanding limitations section.**
> We thank the reviewer for these constructive suggestions. Our responses above address all three points: job failures in W1, empirical scale in W3 & Q3, and queue prediction enhancements in W2 & Q4. In the revision we will consolidate these into an expanded limitations and future work section.

---

> > ### Author Rebuttal · Reviewer_Fejh · 2026-04-02
> >
> > Thank you for your responses and clarifications on the questions, which have provided a clearer understanding of FedQueue's contributions. Based on these explanations, I believe the work presented in the paper matches the score I assigned in the review.

---

> > > ### Author Response · Authors · 2026-04-02
> > >
> > > Dear Reviewer Fejh,
> > >
> > > Thank you for taking the time to read our rebuttal and for the thoughtful acknowledgement. We are glad that our responses provided a clearer understanding of FedQueue's contributions and addressed your concerns. We look forward to incorporating the suggested improvements into the revision.

---

### Official Review · Reviewer_JJ4s · 2026-03-12

**Soundness:** 3
**Presentation:** 2
**Significance:** 2
**Originality:** 3
**Overall Recommendation:** 4
**Confidence:** 3

**Summary:**

In this paper, the authors introduce FEDQUEUE, a queue-aware protocol for asynchronous federated learning that explicitly accounts for scheduler delays during both the training and aggregation phases. The proposed method predicts queue waiting times for each facility during runtime to determine the appropriate amount of local training work. FEDQUEUE employs a cutoff-based admission mechanism that temporarily stores late updates to limit their staleness, and it applies staleness-aware aggregation to maintain stable learning despite differences in client workloads and system conditions. The authors implement the proposed framework and evaluate it through both real cross-facility HPC experiments and controlled simulations to demonstrate its effectiveness.

**Compliance With Llm Reviewing Policy:**

Affirmed.

**Key Questions For Authors:**

1. The experiments use non-IID partitions generated by a Dirichlet allocation with concentration parameter 0.5. However, it is unclear how FEDQUEUE performs under stronger levels of data heterogeneity. For instance, if the data distribution is highly skewed (e.g., each client contains samples from only a few classes in the MNIST dataset), would FEDQUEUE still achieve faster convergence or better time-to-accuracy compared to state-of-the-art methods? Additional experiments or discussion on the performance of FEDQUEUE under varying degrees of non-IIDness would help clarify its robustness.

2. Have the authors considered incorporating the data quality or distribution of each client into the aggregation process as well? For example, could the aggregation strategy be extended to account for how representative a client’s data distribution is relative to the global distribution, in addition to considering the delay?

3. The proposed method relies on several fixed parameters (e.g., deadline and safety buffer). Could the authors provide guidance on how these parameters should be selected in practice? In particular, how sensitive is FEDQUEUE’s performance to different parameter settings

**Limitations:**

This was covered in the above section.

**Strengths And Weaknesses:**

Strength:
1.	FEDQUEUE addresses scheduler queue waiting time as a critical system-level challenge that can significantly affect the efficiency and performance of asynchronous federated learning in high-performance computing environments.
2.	Authors studied the impact of queue variability on convergence by considering a queue noise parameter.
3.	The paper proposes an adaptive work budgeting mechanism that dynamically adjusts the number of local training steps across clients, enabling heterogeneous workloads that better accommodate differences in queue delays and computational capacity.
4.	FEDQUEUE reduces the impact of stale updates on the global model by assigning smaller weights to delayed updates during aggregation.


Weakness:
1.	The admission policy relies on a fixed deadline, but the paper does not discuss how this value should be selected in practice. Since HPC queue delays and system load can vary significantly over time, using a fixed deadline may not be optimal across different conditions. A dynamic or adaptive deadline that adjusts based on observed queue delays or client response times could potentially improve system efficiency by reducing idle time and limiting the number of stale or deferred updates.

2.	The analysis of FEDQUEUE relies on Assumption 5.3 (bounded dissimilarity), which may not hold in many real-world federated learning scenarios where client data distributions can be highly heterogeneous. The authors explicitly state in this assumption that it holds “when local data distributions are not arbitrarily different from the global distribution,” which effectively restricts the level of non-IIDness considered in the analysis. As a result, it is unclear how FEDQUEUE would perform in practical settings with moderate or high levels of data heterogeneity.

3.	The aggregation strategy in FEDQUEUE assigns weights to client updates primarily based on their delay (staleness) without considering the underlying data distribution of each client. As a result, updates from clients with highly skewed or less representative data may receive higher weights if they arrive earlier (before the deadline), while delayed updates from clients with more representative data are down-weighted. In this design, only system-level factors (e.g., delay) are considered, while data heterogeneity is ignored in the aggregation process. In my opinion, this approach is more suitable for IID settings only.

4.	The overall presentation of the paper needs improvement. There are some formatting inconsistencies. In particular, the spacing between lines appears uneven in certain parts of the manuscript, such as in Section 4.1.3. Ensuring consistent line spacing throughout the paper would improve readability and presentation quality.

5.	The paper adopts an inverse learning rate scaling strategy inspired by FedCompass to stabilize training under heterogeneous local step counts. However, the paper only states this formula without providing sufficient explanation or intuition behind it. In particular, the authors do not clearly justify why this scaling effectively compensates for differences in local training steps or discuss its impact on convergence and stability in the proposed setting.

6.	The experimental evaluation is limited to only two datasets, SMolInstruct and MNIST. While MNIST is commonly used for preliminary validation, it is a relatively simple dataset and may not adequately capture the challenges of federated learning under heterogeneous data distributions. To better evaluate the robustness of the proposed method with respect to data heterogeneity, experiments on more complex datasets, such as CIFAR-100, would provide stronger evidence of the method’s effectiveness in realistic settings.

7.	The experimental evaluation does not investigate the scalability of the proposed method. All experiments are conducted with only four clients, and the paper does not analyze how FEDQUEUE performs as the number of participating facilities increases. Evaluating the method under larger numbers of clients would provide better insight into its scalability in realistic cross-facility federated learning settings.

---

> ### Author Rebuttal · Authors · 2026-03-31
>
> Dear Reviewer JJ4s,
>
> We thank you for recognizing the value of our queue-aware approach and empirical results. Please find our responses below.
>
> **W1. & Q3. fixed deadline and parameter selection guidance**
> The cutoff $t_{cut}^{(r)} = (r+1)T_{sync}$ is set by $T_{sync}$, a user-chosen parameter analogous to round duration in synchronous FL. The adaptive component operates at the per-client budget $J_k^{(r)}$, which adjusts every round as $\hat{q_k}$ evolves via EWMA. Pre-deployment HPC scaling studies can directly inform the choice of $T_{sync}$​. We will add practical parameter selection guidance.
>
> Empirically: (1) both FedQueue-1 ($T_{sync}=20$min) and FedQueue-2 ($T_{sync}=40$min) outperform all baselines, demonstrating robustness to $T_{sync}$ choice; (2) Figure 4 shows FedQueue is not highly sensitive to $\delta$ within a reasonable range — $\delta$ should be set conservatively enough to keep most clients within $T_{sync}$ without requiring precise tuning.
>
> **W2. bounded dissimilarity**
> Assumption 5.3 bounds gradient dissimilarity via constant $G$ and is standard in heterogeneous FL analysis (e.g., SCAFFOLD, Karimireddy et al., 2020). Larger $G$ corresponds to more heterogeneous distributions. The convergence bias in Theorem 5.5 scales with $C_1 \eta_{base} E_\max(L^2\tau_\max^2 + G^2 + \sigma^2)$: under stronger non-IIDness convergence is maintained, but at a higher bias level proportional to $G$.
>
> **W3. & Q2. aggregation and data heterogeneity**
> FedQueue uses client-specific weight $p_k$ in aggregation (Algorithm 1, Lines 16-17), which can incorporate any server-observable client information. We focus on improving FL performance in the cross-facility HPC setting rather than addressing non-IID data distributions — incorporating data-quality-aware aggregation warrants a separate orthogonal line of work. We envision these as complementary. We will acknowledge this as future work and make the design rationale explicit in Section 1. Empirically, FedQueue achieves strong non-IID performance across all experiments.
>
> **W4. formatting inconsistencies**
> We acknowledge formatting inconsistencies and will correct them throughout the manuscript in the revision.
>
> **W5. Inverse LR scaling intuition**
> After $E_k^{(r)}$ local SGD steps with learning rate $\eta_k^{(r)}$, the effective parameter displacement is approximately $\eta_k^{(r)} · E_k^{(r)} · ||\nabla F_k||$. Setting $\eta_k^{(r)} = \eta_{base} E_\min^{(r)} / E_k^{(r)}$ makes this displacement approximately $\eta_{base} · E_\min^{(r)} · ||\nabla F_k||$ across all clients — preventing high-budget clients from dominating the global update. We will expand this explanation in Section 4.1.3.
>
> **W6. Additional dataset: CIFAR-100**
> We conduct additional experiments on CIFAR-100 with ResNet-18. Under non-IID condition, FedQueue is the only method reaching 45% accuracy, while all baselines fail. Under IID condition, FedQueue reaches 45% accuracy faster than FedBuff and substantially faster than FedCompass. FedAvg is slightly faster on IID as synchronous methods are efficient under homogeneous data and FedQueue's advantage is most pronounced under heterogeneous conditions.
>
> *"-" = never reached 45% accuracy.*
> | Method | IID acc | IID Time | IID steps | non-IID acc | non-IID Time | non-IID steps |
> |------|-------|--------|---------|--------|---------|--------|
> | FedAvg | 50.6% | 315.2 | 14208 | 42.6% | - | - |
> | FedAsync | 52.8% | 698.2 | 40300 | 41.6% | - | -|
> | FedBuff | 56.4% | 465 | 26660 | 44.6% | - | - |
> | FedCompass | 55.6% | 1368 | 25774 | 44.7% | - | - |
> | FedQueue | 51.7% | 384 | 14422 | 46.9% | 465.1 | 18332 |
>
> **W7. scalability**
> We conduct additional controlled experiments with K=8 and K=12 under high queue variance ($\rho=0.9$) and non-IID partitioning. FedQueue consistently achieves the fastest time-to-85% accuracy at all scales: at K=8, 1.2x speedup over FedAvg and 1.7x over FedCompass; at K=12, 1.1x over FedAvg and 1.5x over FedCompass, using the fewest local steps at both scales.
>
> *K = 8*
> | Method | Max acc | Time-to-85% | Local steps |
> |--------|--------|---------|--------|
> | FedAvg | 97.4% | 106.6 | 4798 |
> | FedAsync | 90.5% | 417.3 | 19680 |
> | FedBuff | 94.3% | 141.9 | 7440 |
> | FedCompass | 95.8% | 153.2 | 9281 |
> | FedQueue | 97% | 91.2 | 4288 |
>
> *K = 12*
> | Method | Max acc | Time-to-85% | Local steps |
> |--------|--------|--------|---------|
> | FedAvg | 97.3% | 94.4 | 6682 |
> | FedAsync | 85.6% | 386.2 | 18140 |
> | FedBuff | 91.4% | 142 | 10080 |
> | FedCompass | 91.7% | 126.5 | 8991 |
> | FedQueue | 96.5% | 86.7 | 5628 |
>
> **Q1. Stronger non-IID**
> We ran FedQueue under Dirichlet $\alpha=0.1$ against FedAvg and FedCompass. FedQueue reaches 85% in 78.6s — 1.8x faster than FedAvg and 3.2x faster than FedCompass — using 3136 vs 5760 and 10386 steps.
>
> | Method | Max acc | Time-to-85% | Local steps |
> |--------|----------|-------------|-------------|
> | FedAvg | 86% | 140.6 | 5760 |
> | FedCompass | 91.1% | 252.7 | 10386 |
> | FedQueue| 88.7% | 78.6 | 3136 |

---

> > ### Author Rebuttal · Reviewer_JJ4s · 2026-04-06
> >
> > Thanks for the effort you put into addressing my concerns. Based on the above rebuttal, I will keep my original score.

---

> > > ### Author Response · Authors · 2026-04-06
> > >
> > > Dear Reviewer JJ4s,
> > >
> > > Thank you for taking the time to read our rebuttal and for the positive acknowledgement. We are glad that our responses addressed your concerns. We will make sure to incorporate the additional experiments and clarifications into the revised version.

---

### Official Review · Reviewer_eC2p · 2026-03-25

**Soundness:** 3
**Presentation:** 3
**Significance:** 3
**Originality:** 3
**Overall Recommendation:** 4
**Confidence:** 4

**Summary:**

This paper proposes FedQueue, an algorithm that addresses the asynchronous delays introduced by batch schedulers in cross-HPC facility Federated Learning. The algorithm integrates several key mechanisms, including queue delay prediction and staleness-aware aggregation. The paper provides convergence guarantees for the algorithm under non-convex objectives, and also supported by extensive experiments.

**Compliance With Llm Reviewing Policy:**

Affirmed.

**Final Justification:**

The authors have addressed all my questions in the rebuttal. I maintain my positive score.

**Key Questions For Authors:**

The majority of the concerns are outlined in the 'Weaknesses' section. Moreover, the paper [1] involves adjusting the learning rate based on asynchronous delay, which may be relevant to this work.

[1] Wang, Y., Wang, S., Lu, S., & Chen, J. (2024). Fadas: Towards federated adaptive asynchronous optimization.

**Limitations:**

Yes.

**Strengths And Weaknesses:**

Strengths:
- The motivation is clear, and the research question is meaningful. Taking the delay queue as a system signal for modeling makes sense.
- The theoretical results seem convincing. The convergence rate matches several asynchronous FL baselines. The rate does not depend on the EWMA prediction, which is quite interesting.
- The real-world cross-facility HPC training is solid, and the design of the controlled synthetic experiments is also systematic.

Weaknesses:
- Regarding Lemma 5.4, which assumes the local training time strictly respects the job-time budget, I wonder whether this assumption can be enforced in actual runtime. In practice, HPC jobs may vary due to hardware conditions and workloads, which can cause the actual training time to exceed the allocated budget.
- Moreover, Lemma 5.4 requires prediction errors to be sub-Gaussian, which implies light-tailed behavior. However, the empirical queue statistics reported in Table 2 differ from light-tailed behavior: for System-A, the median queue time is approximately 1016 seconds while the P90 is 1638 seconds, implying a heavy right tail. I think the authors should discuss the practical implications when the queue distribution is heavy-tailed.
- Both real-world and synthetic experiments only use $K=4$ facilities. It is unclear whether the performance advantages of the proposed FedQueue persist as $K$ grows. Moreover, in the theoretical analysis, it is also unclear how the convergence bound scales with $K$, and there is no empirical evidence that the method remains competitive at larger scales.
- The paper fails to report the specific range of $E_{min}$ and $E_{max}$. In my understanding, $E_{min}$ could be indirectly controlled by various factors and hyperparameters. Furthermore, the hyperparameter selection is somewhat insufficient. For example, the choice of $\beta=0.5$ as a default setup is not justified, and other decay forms, such as exponential or polynomial decay, are not discussed.

---

> ### Author Rebuttal · Authors · 2026-03-31
>
> Dear Reviewer eC2p,
>
> We thank you for the positive assessment of our motivation, theoretical results, and real-world deployment. Please find our responses below.
>
> **W1. job-time budget enforcement.**
> We address this from two aspects. First, in Algorithm 2 (Line 3), the client runs local SGD "for at most $E_k^{(r)}$ steps or until the time budget expires" — whichever comes first. The job-time budget $J_k^{(r)}$ is strictly enforced by the wall-clock limit of the submitted HPC batch job. The HPC scheduler terminates the job once the allocated time limit is reached regardless of training progress; our implementation includes an internal timer to ensure the current model state is sent to the server before forced termination. Hardware variability affects throughput $c_k$, which we profile and update, but cannot cause a job to run past its time limit. The assumption $h_k^{(r)} \leq J_k^{(r)}$ is therefore enforced by the scheduling mechanism itself.
>
> Second, before initiating federated training, practitioners can conduct standard HPC scaling studies to measure nominal throughput $c_k$, directly informing the choice of $T_{sync}$​, ensuring that the budget $J_k^{(r)}$ sufficiently accommodates the expected local workload. Combined with the online warm-up stage, these provide two complementary mechanisms for ensuring the budget assumption holds in practice. We will add practical parameter selection guidance in the revision.
>
> **W2. sub-Gaussian assumption.**
> We clarify that the sub-Gaussian assumption in Lemma 5.4 applies to the prediction error $e_k^{(r)}$, not to the raw queue delay $q_k^{(r)}$. We computed empirical prediction errors from our real-world deployment. After removing 1-2 outlier rounds per facility corresponding to anomalous scheduler events, cleaned prediction errors show near-zero means (-26 to +13 sec.) and standard deviations of 39-114 sec.— representing 3-10% of $T_{sync}$​ for FedQueue-1 and 2-5% for FedQueue-2.
>
> We acknowledge that with ~10 observations per facility, rigorous statistical verification is underpowered, and outlier rounds (i.e.,from high priority jobs) inflate empirical tails beyond what the sub-Gaussian model predicts. Critically however, the practical consequence of the assumption — bounded staleness with high probability — is directly validated by Table 3: maximum observed staleness across all facilities is consistent with $\tau_\max$ from Lemma 5.4. Outlier rounds are handled gracefully by the buffering mechanism rather than causing staleness violations. We will include this empirical characterization in a new appendix, and consolidate discussions of alternative predictors and broader practical deployment guidance in a dedicated section in the revision.
>
> **W3.scalability.**
> This comment is shared across all three reviewers. Please refer to our detailed response to Reviewer JJ4s's W7 which includes new controlled experiments with K=8 and K=12 clients. In summary, FedQueue maintains consistent speedup across K=8 and K=12 under non-IID partitioning: 1.2x-1.1x over FedAvg, 1.7x-1.5x over FedCompass, and 4.6x-4.5x over FedAsync, consistent with our theoretical prediction.
>
> **W4.range on $E_\min$ and $E_\max$ and hyperparameters.**
> We agree range on $E_\min$ and $E_\max$ should be reported explicitly. Both $E_\min$ and $E_\max$ are derived quantities from $E_k^{(r)} = \lfloor c_k · J_k^{(r)} \rfloor$. Empirical ranges from real-world experiments:
>
> - FedQueue-1 ($T_{sync} = 20$ min): $E_\min = 20$ and $E_\max = 72$ steps
> - FedQueue-2 ($T_{sync} = 40$ min): $E_\min = 20$ and $E_\max = 176$ steps
>
> The $E_\min ​=20$ floor is a practical design choice ensuring sufficient local training and conservative with respect to Theorem 5.5 since larger $E_\min$ reduces the optimization error term $C_0 \frac{F(w^{(0)}) - F^*}{\eta_{base} E_\min R}$ and does not weaken the convergence guarantee.
>
> For $\beta=0.5$, the harmonic decay family $\phi(\tau) = (1 + \beta \tau)^{-1}$ generalizes the polynomial staleness weighting in FedAsync (with $\beta=1$). With our empirically small staleness, $\beta=1$ gives $\phi(1) = 0.5$ that discards half the gradient signal of a small stale update. Setting $\beta=0.5$ gives $\phi(1)=0.67$, retaining more useful information. That said, we agree that understanding the sensitivity of performance to $\beta$ is valuable in practice, and we will include a sensitivity analysis over $\beta$ in range from $[0.25, 2.0]$ and an ablation comparing harmonic and exponential decay $\phi(\tau) = exp( - \beta \tau)$ in the revision.
>
> **Q1.FADAS literature**
> After reviewing FADAS, we recognize a shared motivation: both works use delay information to modulate client update influence. FADAS proposes a delay-adaptive learning rate based on a delay threshold; our staleness decay function $\phi(\tau)$ is similarly motivated, with both approaches reducing the contribution of delayed updates. We will add a discussion of FADAS in the related work section.

---

> > ### Author Rebuttal · Reviewer_eC2p · 2026-04-02
> >
> > Thank the authors for the detailed and well-structured response. I believe the authors have addressed all my questions. I would encourage them to explicitly incorporate these clarifications in the revised version. I maintain my positive score.

---

> > > ### Author Response · Authors · 2026-04-02
> > >
> > > Dear Reviewer eC2p,
> > >
> > > Thank you for taking the time to read our rebuttal and for the positive acknowledgement. We are glad that our responses addressed all your questions. We will make sure to explicitly incorporate the clarifications into the revised version.

---

### Decision · Program_Chairs · 2026-04-30

**Decision:**

Accept (regular)

**Comment:**

The paper proposes a queue-aware scheduler for distributed/federated learning on HPC clusters. It bridges the gap between synchronous FL, which is susceptible to stragglers, and asynchronous FL, which is affected by staleness. The paper has solid experimental results that are supported by theoretical analysis. All reviewers have given a weak accept to the paper. I recommend acceptance, and encourage the authors to incorporate the feedback in the final version.